# Computational modelling of the impact of anatomical changes on ECGs in left ventricular hypertrophy

Mohammadreza Kariman[1], Karli Gillette[1,2,3] , Matthias A. F. Gsell[1] , Anton J. Prassl[1] ,
Gernot Plank[1,2] and Christoph M. Augustin[1,2]

[1] *Gottfried Schatz Research Center: Division of Medical Physics and Biophysics, Medical University of Graz, Graz, Austria*
[2] *BioTechMed-Graz, Graz, Austria*
[3] *Department of Biomedical Engineering, University of Utah, Salt Lake City, UT, USA*

Handling Editors: Eleonora Grandi & Brian Delisle

The peer review history is available in the Supporting Information section of this article
(https://doi.org/10.1113/JP287954#support-information-section).

**Abstract figure legend** Eccentric and concentric left ventricular hypertrophy distinctly alter 12-lead ECG signals. Eccentric hypertrophy predominantly increases voltages in the precordial leads, whereas concentric hypertrophy affects voltages across all leads in a variable manner. Both forms of hypertrophy are associated with a prolongation of the QRS duration by up to 40%.

**Mohammadreza Kariman** is a researcher in the Computational Cardiology Lab at the Medical University of Graz. He earned a BSc in Biomedical Engineering from the University of Birmingham and an MSc in Biomechanics from the University of Manchester, where he investigated temperature effects on cardiac electrophysiology. He later conducted research at Imperial College London, examining how aortic geometry influences atherosclerosis using computational fluid dynamics. His current work focuses on developing and calibrating digital twin models of cardiac electromechanics, with a broader interest in advancing personalised heart modelling for clinical use.

**Abstract** Left ventricular hypertrophy (LVH) is characterised by an increase in the mass and volume of the left ventricle, typically manifested as ventricular wall thickening and/or dilation. Due to its potential to cause severe, life-threatening complications, ongoing research continues to explore its underlying mechanisms. This study aimed to determine how wall thickening and dilation specifically impact ECG waveforms, isolating these anatomical alterations without considering potential electrophysiological changes associated with LVH – a scenario achievable only through computational modelling. To accomplish this, eccentric and concentric cardiac models – with growth levels from 10% to 100% mass increase – were generated using a kinematic growth, finite element model derived from a healthy control model. Activation sequences were simulated for each model using a pseudo-bidomain reaction-eikonal approach, and 12-lead ECGs were recorded from the hypertrophy models and compared to the control. Results indicated that activation patterns in eccentric hypertrophy models resembled the healthy model, while concentric hypertrophy models displayed substantial deviations. Both types of hypertrophy types led to prolonged QRS durations by up to 21 ms – a 40% increase from baseline – even in the absence of electrical remodelling. Eccentric hypertrophy increased amplitudes in precordial leads, minimally affecting limb leads, while concentric hypertrophy impacted all 12 leads with varied amplitude changes. Leads aVL, V1 and V5/V6 emerged as the most sensitive to anatomical changes. These findings could enhance the accuracy of LVH diagnosis using ECGs, offering a cost-effective strategy to complement clinical evaluation and imaging, ultimately improving LVH detection and management.

(Received 31 October 2024; accepted after revision 17 July 2025; first published online 2 September 2025)

**Corresponding author** C. M. Augustin: Gottfried Schatz Research Centre: Division of Medical, Physics and Biophysics, Medical University of Graz, Neue Stiftingtalstraße 6 (MC2.H.)/III, 8010 Graz, Austria. Email: christoph.augustin@medunigraz.at

## Key points

- Computational simulations revealed distinct effects of anatomical changes in eccentric and concentric left ventricular hypertrophy on 12-lead ECG signals.
- Eccentric hypertrophy primarily affected the precordial leads, showing notable voltage amplitude increases across all precordial lead measurements.
- Concentric hypertrophy affected all 12 leads without a clear pattern of amplitude change, displaying both increases and decreases.
- Both eccentric and concentric hypertrophy resulted in a consistent prolongation of the QRS complex, showing up to 40% increase from baseline, even in the absence of electrophysiological remodelling.
- Leads aVL, III, V1 and V5/V6 were identified as the most sensitive to LVH, with computational results aligning well with independent clinical measurements.

## Introduction

Left ventricular hypertrophy (LVH) is a condition manifested by an increase in the left ventricular (LV) mass, marked by thickening and/or dilation of the LV wall of the heart (Tomaselli & Marbán, 1999). LVH is a physiological adaptation that typically occurs in response to prolonged pressure or volume overload on the heart, stemming from intense physical activity or pathological conditions such as hypertension or aortic valve stenosis (Grossman & Paulus, 2013; Paar et al., 2023). The thickening of the ventricular wall is a compensatory mechanism to enhance contractility and sustain cardiac output. LVH can manifest in either an adaptive or a maladaptive pattern. In the early stages, adaptation acts as a compensatory and beneficial mechanism. However, if pathological stimuli persist, maladaptation becomes more prominent, featuring detrimental mechanisms that promote cell death and fibrosis (Lips, 2003; Niestrawska et al., 2020). Cardiomyocyte degeneration results in significant alterations in ventricular tissue architecture, particularly the development of fibrosis (Fielitz et al., 2001). The presence of fibrosis is considered a pivotal factor in the transition from compensated hypertrophy to

a maladaptive state, as it is associated with an increased likelihood of re-entrant arrhythmias (Opie et al., 2006; Wolk, 2000).

LVH can be classified into two distinct types: concentric and eccentric hypertrophy (Khouri et al., 2010; Peters, 1996). Concentric hypertrophy is characterised by an increased LV mass with an elevated ratio of LV wall thickness to radius ($h/R$) (Tomaselli & Marbán, 1999). Two main forms are recognised: symmetric concentric hypertrophy, characterised by an almost equal increase in thickness across the entire LV and septum, and asymmetric concentric hypertrophy, in which the thickening is non-uniform and predominantly affects either the free wall or the septum, with the most frequent variant being asymmetric septal hypertrophy (Basso et al., 2021). This type typically occurs due to pressure overload in conditions like aortic stenosis, where the ventricular myocardium undergoes elevated wall stress (Gerdes, 2002; Grossman et al., 1975). In such cases, concentric hypertrophy functions to mitigate the increased systolic pressures in the LV. This is achieved by replicating sarcomeres in parallel in response to peak wall stress, eventually resulting in increased wall thickness and restoring wall stress toward normal levels (Grossman et al., 1975). Eccentric hypertrophy, induced by volume overload conditions like mitral or aortic valve regurgitation, leads to a serial deposition of sarcomeres in response to increased end-diastolic wall stress (Grossman & Paulus, 2013). This adaptive mechanism maintains the $h/R$ ratio, resulting in ventricular dilation to accommodate a larger blood volume and ultimately reduce diastolic wall stress (Bornstein et al., 2024).

Considering the significance of LVH, several diagnostic methods are available for detecting this life-threatening cardiac condition. Echocardiography is widely used (Lang et al., 2015) for its real-time imaging capabilities and cost-effectiveness, while cardiac magnetic resonance imaging (MRI) is considered the gold standard for diagnosing LVH due to its detailed anatomical visualisation and ability for myocardial tissue characterisation (Lewis & Rider, 2020). Both modalities allow estimation of LV wall thickness and mass, which are used to establish criteria for assessing the prevalence of LVH by indexing to body surface area (BSA) or height (Cuspidi et al., 2009). Despite advancements in imaging, 12-lead electrocardiography (ECG) remains essential in both research and clinical settings and is often the initial diagnostic tool for identifying LVH due to its accessibility and affordability. As such, improving ECG criteria and the detection of LVH using ECGs is a crucial area of ongoing research, with a strong need for refinement to increase diagnostic accuracy. Traditional ECG criteria can sometimes lead to misdiagnosis, as similar patterns appear in various cardiac conditions. For instance, recent studies indicate that approximately

one-third of patients diagnosed with left bundle branch block (LBBB) based on traditional ECG criteria may actually have inter-ventricular conduction delay due to LVH (Strauss, 2012; Strauss et al., 2011). Over the past century, more than 30 ECG criteria have been proposed for diagnosing LVH, each with unique performance characteristics (Hancock et al., 2009). While these criteria generally exhibit high specificity, they often have low sensitivity (Krittayaphong et al., 2013) and show a varying performance, partly due to factors such as sex, body size and ethnicity (Alfakih et al., 2004; Casale et al., 1985; Chapman, 1999; Hancock et al., 2009; Salamaga et al., 2023) or geographic location and patient cohorts (Hancock et al., 2009; Song et al., 2021; Su et al., 2017). Among the criteria, the Sokolow–Lyon (Sokolow & Lyon, 1949) and Cornell (Casale et al., 1985) criteria have been extensively utilised over the past decades. Introduced in 2017, the relatively novel Peguero–Lo Presti criterion has demonstrated improved sensitivity and overall accuracy in ECG-based LVH diagnosis (Marcato et al., 2022; Peguero et al., 2017). In recent years, machine learning (ML) tools have gained momentum to diagnose LVH from ECGs (Jothiramalingam et al., 2021; Liu et al., 2023; Sammani et al., 2022). While many of these tools show improvements in sensitivity, most have fallen short to outperform specificity of classic ECG criteria (Rabkin, 2024).

To address the overall diagnostic challenges, particularly the low sensitivity of conventional ECG criteria, we studied the impact of anatomical changes in LVH through computational simulations. This approach allowed us to isolate these alterations for a more targeted investigation without confounding effects of changes in electrical activity. By comparing the simulated activation sequences, body surface potential maps (BSPMs) and ECGs from LVH models to a control model calibrated on patient-specific ECG data, we were able to analyse mechanistic changes in both eccentric and concentric LVH – an approach that cannot be achieved through ML or clinical studies alone.

To simulate LVH using a healthy control model, the concept of kinematic growth was applied (see 'Methods' section). This method, which facilitates volumetric growth within a continuum mechanics framework, was first introduced by Rodriguez et al. (1994), based on the principles of plasticity. In this model, growth is characterised as changes in the shape and size of an unloaded body through inelastic deformation. These stress-free changes, caused by the addition or loss of mass, create an intermediate, incompatible configuration. To restore geometric compatibility, elastic deformation is applied, organising the volumetric elements into an unloaded body and generating residual stresses. The idea of kinematic growth has been applied in several studies to patient-specific heart geometries (e.g. Göktepe et al., 2010;

Kroon et al., 2009; Rausch et al., 2017). Results of electro-physiological (EP) simulations using these LVH models (see 'Results' section) revealed that activation patterns in eccentric hypertrophy models were similar to the healthy model, while concentric hypertrophy showed significant deviations. Both types of hypertrophy prolonged QRS durations even without electrical remodelling. Eccentric hypertrophy increased amplitudes primarily in precordial leads, whereas concentric hypertrophy affected all 12 leads, with aVL, V1 and V5/V6 being the most sensitive to anatomical changes in both types. Mechanistic differences in the context of LVH are discussed in great detail in the 'Discussion' section. Additionally, to strengthen the credibility of our results, we conducted an independent analysis of ECGs from a large, publicly available data-set (Wagner et al., 2020). This comprehensive resource offered a robust platform for cross-verifying our findings, ensuring that our conclusions are both reliable and generalisable across a broader patient population.

Our study represents a novel application of kinematic growth models to examine factors influencing ECG patterns, allowing for a focused assessment of anatomical alterations in isolation. Specifically, we aimed to determine whether changes like increased LV mass and volume are primary drivers of ECG variations in LVH. We expect that our findings will enhance our understanding of the complex mechanisms underlying LVH, enabling us to refine diagnostic criteria and ultimately improve the precision of ECGs in differentiating LVH from other cardiac conditions, thereby enhancing overall diagnostic accuracy.

## Methods

### Ethical approval

A detailed anatomical model of a healthy human heart, along with a reconstruction of the subject's torso, was generated using magnetic resonance (MR) images from a 45-year-old male subject. This MRI study received approval from the Ethical Review Board of the Medical University of Graz (EKNr: 24–126 ex 11/12), and participants provided written informed consent. For further details on image acquisition, see Gillette, Gsell, Prassl, et al. (2021).

### Anatomical model

The heart model was registered within the torso to preserve its anatomical position, including its natural orientation and rotation using an iterative closest point algorithm (Besl & McKay, 1992), and an anatomical mesh was generated. The blood pools in the heart's chambers were labelled separately, while the rest of the torso was treated as a homogeneous material, with no separate labelling for lungs, ribs or other internal anatomical features. The target mesh resolution was 1200 μm for the four-chamber heart and 2260 μm for the torso, striking a balance between computational efficiency and accurate representation of cardiac structures.

Fibre architecture within the ventricles was generated using a rule-based method as described by Bayer et al. (2012), with longitudinal fibre angles rotating from 60° to −60° and transverse fibre angles rotating from −65° to 25°, from the endocaridal to the epicardial surface. For the atria, fibres were incorporated from an end-ocardial fibre atlas following Roney et al. (2021) and extended transmurally to the volumetric atrial mesh using a kd-tree (Gillette et al., 2022). Mesh generation and fibre integration were performed in CARPentry Studio (NumeriCor GmbH, Graz, Austria).

### Biomechanical model

To simulate LVH using the healthy control model from the Methods section 'Anatomical model' a kinematic growth approach was used which requires two key components: a strain–energy function to model the hyperelastic material behaviour and an evolution equation for the growth tensor. For the strain–energy function we assumed the cardiac tissue to be a hyperelastic, nearly incompressible and anisotropic material with a non-linear stress–strain relationship (Holzapfel & Ogden, 2009). Ventricular material parameters were based on Gültekin et al. (2016) for the LV and we employed slightly elevated material stiffness for the right ventricle (RV) (Sacks & Chuong, 1993). Atrial parameters were based on Augustin et al. (2020) where overall stiffness in the atria was increased by a scaling factor. This adjustment enhanced the stability of the kinematic growth algorithm while limiting growth in the atrial regions, which were not our primary focus. More details on the passive mechanical model are given in Appendix A including a list of material parameters in Table A1.

To apply boundary conditions, the surface of the four chamber model was decomposed in three parts: the end-ocardium, the epicardium, and the inlets and outlets of the vessels attached to the heart. To constrain cardiac motion at the vessels, omni-directional, spring-type boundary conditions were implemented (Land & Niederer, 2018). Normal stress boundary conditions were applied to the LV endocardium to simulate intracavitary pressure loading. To isolate the effects of LVH, the remaining heart chambers were not pressurised but instead stabilised using omni-directional spring boundary conditions. To account for the mechanical constraints imposed by the pericardium, we applied spring boundary conditions with stiffness values varied for each growth scenario. For eccentric growth, a relatively compliant value of

0.005 kPa/μm was used, allowing more expansion. For symmetric eccentric growth, a slightly stiffer value of 0.01 kPa/μm was applied. For asymmetric eccentric growth, a value of 0.3 kPa/μm was imposed, restricting LV expansion and fibre stretch in the LV free wall and thus promoting septal thickening relative to the symmetric eccentric growth case.

For the evolution equation for the growth tensor, we followed the approach outlined by Genet et al. (2016), which distinguishes between eccentric growth, driven by serial sarcomere deposition, and concentric growth, driven by parallel sarcomere deposition. Growth kinetics were assumed to be stretch-driven in both the transverse and longitudinal directions. To regulate excessive tissue growth locally, we introduced a sigmoid function (see Appendix B) to scale the kinematic growth rate, thereby limiting the maximum extent of growth. Kinematic growth was triggered when local fibre stretch exceeded a critical value $\lambda_{crit}$, which was calculated as a regionally varying baseline stretch under physiological conditions, which corresponds to a passive inflation experiment using a pressure of $p_{inflate} = 5$ mmHg within the cavity of the LV. In the kinematic growth experiment, we then inflated the model to a pressure of $p_{overload} = 10$ mmHg over a duration of $t_{increase} = 100$ time units. We maintained this pressure for $t_{growth} = 600$ time units, after which we gradually reduced the pressure to 0 mmHg over $t_{decrease} = 100$ time units. This process models growth within a normalised time interval, which corresponds to a physical time frame of several months to years. We repeated this procedure until we exceeded a specified threshold of mass increase. Here, thresholds ranged from 10% to 100%, applied in 10% increments, as shown in Table 1. The passive inflation procedure to compute $\lambda_{crit}$ was repeated after each kinematic growth experiment. This process ultimately produced 30 models of cardiac hypertrophy, with 10 models for each type of hypertrophy. See Appendix B for equations and more details on the kinematic growth model and Table A1 for the parameters.

Particularly in the more extreme cases of concentric growth, mesh intersections and irregular endocardial surfaces were observed. These irregularities were corrected and smoothed using *meshtool* (Neic et al., 2020) and *CARPentry Studio* (NumeriCor GmbH, Graz, Austria) to ensure mesh integrity for the EP simulations in the Methods section 'Electrophysiology model'. In this step, care was taken to maintain the overall shape and LV mass. Afterwards, the four-chamber geometries were embedded into the torso model, and the labelling of blood pools and torso meshing, as described above in the Methods section 'Anatomical model', was repeated. Special attention was given to ensuring that the hypertrophic four-chamber hearts were in the same position as the control model, maintaining consistency between the EP simulations.

Kinematic growth and EP simulations, described in the following section ('Electrophysiology model'), were performed using the finite element framework *Cardiac Arrhythmia Research Package* (*CARPentry*) (Augustin et al., 2016; Neic et al., 2017), built upon extensions of the *openCARP* framework (Plank et al., 2021) (http://www.opencarp.org). The source code of *openCARP* is public and the software is freely available for academic use, with additional extensions provided upon request. Mesh generation, manipulation and measurements were performed using *meshtool* (Neic et al., 2020), an open-source tool available at https://bitbucket.org/aneic/meshtool, and *CARPentry studio* distributed by NumeriCor GmbH, Graz, Austria (https://numericor.at).

## Electrophysiology model

The simulations of current flow within both intra- and extracellular domains were achieved using a pseudo-bidomain reaction-eikonal (R-E$^+$) model (Neic et al., 2017) and activation sequences, ECG traces and $\Phi_e$ distributions within the torso surrounding the heart were computed.

Compared to the full bidomain model, the pseudo-bidomain R-E$^+$ model is associated with a lower computational cost, making it a more efficient option for large-scale simulations while essential accuracy is retained even for coarser meshes with resolutions as chosen in the Methods section 'Anatomical model'.

The Ten Tusscher ionic model, which is a detailed human ventricular cellular model, was used for the ventricles (Ten Tusscher & Panfilov, 2006). Since the focus was solely on ventricular depolarisation, the atrial structures were not electrically active in the EP simulation. The conduction velocities in the ventricles were set at 0.6 m/s along the myocardial fibre direction, 0.4 m/s along the transverse direction, and 0.2 m/s along the normal direction (Clerc, 1976). The intracellular conductivities were set to 0.34 S/m for $\sigma_{il}$, and 0.6 S/m for both $\sigma_{it}$ and $\sigma_{in}$ (Roberts & Scher, 1982). Extracellular conductivities were defined as 0.12 S/m for $\sigma_{el}$, and 0.8 S/m for both $\sigma_{et}$ and $\sigma_{en}$ (Roberts & Scher, 1982).

To facilitate ventricular activation, a simplified representation of the His–Purkinje system was modelled based on the sites of earliest activation (Gillette, Gsell, Bouyssier, et al., 2021). The model featured a fast-conducting sub-endocardial layer, encompassing approximately 80% of the endocardial layer, extending from 10% above the apex to 10% below the base in the apico-basal direction, with an isotropic conduction velocity set to 2 m/s to replicate the complex network of Purkinje fibres within the sub-endocardium. On the LV sub-endocardial surface, three fascicles were designated

**Table 1. LV myocardial volume, LV mass, mass index and hypertrophy status for cardiac models under eccentric and concentric growth conditions**

| Cardiac model | Tissue volume (ml) | Mass (g) | Mass index (g m$^{-2.7}$) | Hypertrophy |
|---|---|---|---|---|
| Healthy | 211.88 | 223.11 | 38.34 | False |
| E 10% | 234.79 | 247.23 | 42.48 | False |
| E 20% | 257.11 | 270.74 | 46.52 | False |
| E 30% | 277.18 | 291.87 | 50.15 | False |
| E 40% | 298.48 | 314.30 | 54.00 | True |
| E 50% | 319.78 | 336.73 | 57.86 | True |
| E 60% | 342.38 | 360.53 | 61.95 | True |
| E 70% | 361.01 | 380.15 | 65.32 | True |
| E 80% | 382.60 | 402.88 | 69.23 | True |
| E 90% | 400.70 | 421.93 | 72.50 | True |
| E 100% | 421.94 | 444.30 | 76.34 | True |
| AC 10% | 238.89 | 251.55 | 43.22 | False |
| AC 20% | 255.48 | 269.02 | 46.22 | False |
| AC 30% | 268.89 | 283.14 | 48.65 | False |
| AC 40% | 284.22 | 299.28 | 51.42 | True |
| AC 50% | 308.77 | 325.13 | 55.87 | True |
| AC 60% | 336.32 | 354.15 | 60.85 | True |
| AC 70% | 373.18 | 448.32 | 64.12 | True |
| AC 80% | 405.73 | 487.41 | 69.71 | True |
| AC 90% | 440.82 | 529.57 | 75.74 | True |
| AC 100% | 451.71 | 542.66 | 77.62 | True |
| SC 10% | 235.29 | 247.76 | 42.57 | False |
| SC 20% | 255.77 | 269.33 | 46.28 | False |
| SC 30% | 274.98 | 289.56 | 49.75 | False |
| SC 40% | 297.00 | 312.75 | 53.74 | True |
| SC 50% | 317.35 | 334.16 | 57.42 | True |
| SC 60% | 341.80 | 359.91 | 61.84 | True |
| SC 70% | 360.31 | 379.41 | 65.19 | True |
| SC 80% | 381.41 | 401.62 | 69.01 | True |
| SC 90% | 405.67 | 427.17 | 73.40 | True |
| SC 100% | 424.53 | 447.03 | 76.81 | True |

Mass indices are calculated using height-based indexing criteria for hypertrophy identification. The control cardiac model corresponds to an individual with a height of 1.92 m. LVH is diagnosed when mass index exceeds 51 g m$^{-2.7}$. Myocardial density was assumed to be 1.053 g/ml for the calculation of ventricular mass (Vinnakota & Bassingthwaighte, 2004). E, eccentric; AC, asymmetric concentric; SC, symmetric concentric.

for the anterior, posterior and septal fascicles, while the RV sub-endocardial surface was equipped with two fascicles to account for the moderator band and septal fascicles (Durrer et al., 1970). Conductivity in the blood pools and general tissue in the torso volume conductor were set to the nominal values 0.7 S/m and 0.22 S/m, respectively (Keller et al., 2010). Further details on the EP model can be found in the prior work conducted by Gillette et al. (2022).

### Localisation of earliest activation sites

The earliest activation sites, i.e. root fascicles, for the healthy model were obtained using methods outlined in Gillette, Gsell, Prassl, et al. (2021). To accurately map these sites on the sub-endocardial layer of healthy ventricular models onto hypertrophy models while maintaining consistency in endocardial activation regions, a Python algorithm – see also Appendix C for more information – was developed that leveraged concepts from the universal ventricular coordinates (UVCs) proposed by Bayer et al. (2018). UVCs have proven instrumental in facilitating the mapping of structures of interest between ventricular meshes, effectively functioning as a global positioning system for ventricular heart geometries.

The positions of the fascicles were verified by comparing them with their locations after applying the kinematic growth algorithm. In the control model, fascicles were tagged with material markers, allowing their positions to be tracked throughout the growth procedure. Final positions were then compared with

the mapped fascicles, showing good agreement. It is important to note that the final positions after growth were not used, as fascicle size changed and locations within the ventricular wall were observed. While these changes might be a consequence of LVH leading to abnormal electrical activation (Lyon et al., 2018), this study focuses solely on anatomical alterations, excluding changes in the conduction system. Thus, the mapping procedure based on UVCs was preferred.

### Activation sequence and body surface potential maps

Local activation times (LATs) derived from transmembrane voltage distributions were used to reconstruct isochronal maps that depict the sequence of myocardial tissue activation during depolarisation. Activation patterns in both control and hypertrophic cardiac models were visualised side-by-side, illustrating how structural changes, such as dilation and wall thickening, alter the activation sequence independently of electrophysiological remodelling.

BSPMs were generated and sectioned in both transverse and coronal planes. This approach created a three-dimensional view of potential distribution across the torso during ventricular depolarisation, which allowed for a detailed comparison between the healthy control model and the LVH models. Through this method, we were able to explore the relationship between altered cardiac structure and its electrical manifestations on the body's surface.

### LV mass indexation

LV mass is a critical biomarker for image-based diagnosis of LVH. In clinical practice, precise measurements of LV mass are reliably obtained using CT, MRI and echocardiography (Celebi et al., 2010; Lang et al., 2015). LV mass is a dynamic parameter that varies according to individual factors like body size, sex and ethnicity. Therefore, to accurately diagnose LVH, LV mass measurements must be adjusted to account for these variations (Chirinos et al., 2010; De Simone et al., 1992). To facilitate this, several indices have been developed over the past decades. Among the available indices, two stand out as particularly useful: one adjusts LV mass relative to BSA, and the other adjusts it based on height (De Simone et al., 1992; Devereux et al., 1984). While clinical diagnosis has traditionally relied on indexing LV mass by BSA, recent studies suggest that indexing LV mass by height offers greater accuracy (Cuspidi et al., 2009). A study involving 2213 patients demonstrated substantial differences in LVH prevalence, with LV mass indexed to BSA yielding a prevalence of 31.0% compared to 46.5% when indexed to height (Cuspidi et al., 2009). Notably,

indexing LV mass to BSA tends to underestimate LVH prevalence, particularly in obese and overweight individuals.

In this paper, to classify hypertrophied models, we employed the method of indexing LV mass to height. This involves dividing LV mass (in grams) by height (in metres) raised to the power of 2.7. In male adults, LVH is diagnosed when these indexed values exceed 51 g m$^{-2.7}$ (de Simone et al., 1995).

### Baseline ECGs

A 12-lead ECG was recorded from the individual corresponding to the healthy cardiac model. The recorded ECG was filtered using a 150 Hz low-pass filter, a 0.5 Hz high-pass filter and a 50 Hz bandstop filter to eliminate electrical noise. Anatomical sites for electrode positions were recovered from the MRI-compatible electrodes to ensure consistency with the measured ECG when constructing the simulated ECG. In the EP settings, the anatomical locations of the fascicles on the LV and RV sub-endocardial layers were adjusted to replicate the ECG morphology of the measured signal, a process known as ECG personalisation. This ECG personalisation methodology was performed in accordance with Gillette, Gsell, Prassl, et al. (2021) and Gillette et al. (2022). A scaling factor of 0.28 was applied to the simulated ECG to align the signals with the peak R-wave amplitude. This adjustment was necessary due to the elimination of lungs and ribs in the simulation to accommodate the larger hypertrophied hearts, which would otherwise conflict spatially with these structures. Additionally, skin and fat layers – which would act as resistive layers between the heart and the electrodes on the body surface – were excluded as they were not part of the personalisation in the original paper of Gillette, Gsell, Prassl, et al. (2021). The simulated ECG of the control cardiac model served as the baseline for comparing ECGs from hypertrophy models, allowing for the identification of morphological differences due to anatomical changes associated with LVH.

### LVH ECG criteria

Of the numerous ECG criteria that have been proposed for diagnosing LVH over the past century (Hancock et al., 2009), the Sokolow–Lyon (Sokolow & Lyon, 1949) and Cornell (Casale et al., 1985) criteria have been most widely utilised. Recently, the Peguero–Lo Presti criterion has been introduced, demonstrating improved sensitivity and accuracy in ECG-based LVH diagnosis by measuring the amplitude of the deepest S-wave in conjunction with the S-wave amplitude of lead V4 (Marcato et al., 2022; Peguero et al., 2017).

These three ECG criteria, illustrated in Fig. 1, were

applied to assess the ECGs obtained from EP simulations of LVH cardiac models using an automated algorithm to evaluate R- and S-wave amplitudes in the QRS segment.

## Analysis of clinical ECGs

The PTB-XL ECG dataset from PhysioNet (Wagner et al., 2020), a large publicly available resource, was used to analyse key differences in the QRS segment, focusing specifically on the variations in maximum amplitude between individuals with LVH and those with normal cardiac function. We randomly selected 4000 records and removed those where the QRS complex could not be reliably detected. This resulted in a total of 3937 records: $N_{ctrl} = 1967$ normal control patients (983 female and 984 male with normal ECGs) and $N_{LVH} = 1970$ LVH patients (989 female and 981 male with ECGs showing LVH). Raw ECG signals were pre-processed using a 150 Hz low-pass filter, a 0.5 Hz high-pass filter and a 50 Hz band-stop filter to mitigate electrical noise. The QRS segment was extracted from each 12-lead ECG, and the maximum amplitude – irrespective of polarity – was detected for each lead. The average of the maximum amplitudes across all leads was then computed for each cohort (see eqns (1) and (2)) with a 95% truncated mean, i.e. the mean is computed by removing the lowest and highest 2.5% of values from the data distribution and then calculating the arithmetic mean of the remaining values. This approach mitigates the impact of outliers on the mean estimation. Finally, the average maximum amplitude in

the LVH cohort was compared to that of the normal cohort eqn (3):

$$\overline{\max\left(\text{Lead}_i^{ctrl}\right)} = \frac{1}{|N_{ctrl}|} \sum_{j=1}^{N_{ctrl}} \left(\text{Lead}_{i,j}^{ctrl}\right), \quad (1)$$

$$\overline{\max\left(\text{Lead}_i^{LVH}\right)} = \frac{1}{|N_{LVH}|} \sum_{j=1}^{N_{LVH}} \left(\text{Lead}_{i,j}^{LVH}\right), \quad (2)$$

$$D_i = \frac{\overline{\max\left(\text{Lead}_i^{LVH}\right)} - \overline{\max\left(\text{Lead}_i^{ctrl}\right)}}{\overline{\max\left(\text{Lead}_i^{ctrl}\right)}}, \quad (3)$$

where $i \in$ {aVL, I, aVR, II, aVF, III, V1, V2, V3, V4, V5, V6} is the lead number, $j$ is the patient number, and $D_i$ is the relative difference in amplitude between the normal and the LVH cohorts.

## Results

### Models of eccentric and concentric hypertrophy

Table 1 and Fig. 2 present the results of the kinematic growth algorithm described in the Methods section 'Biomechanical model' for serial and parallel sarcomere deposition. The computational method yielded predictions of eccentric and concentric hypertrophy resulting from LV overload. The kinematic growth procedure was iteratively executed until a specific threshold, given in

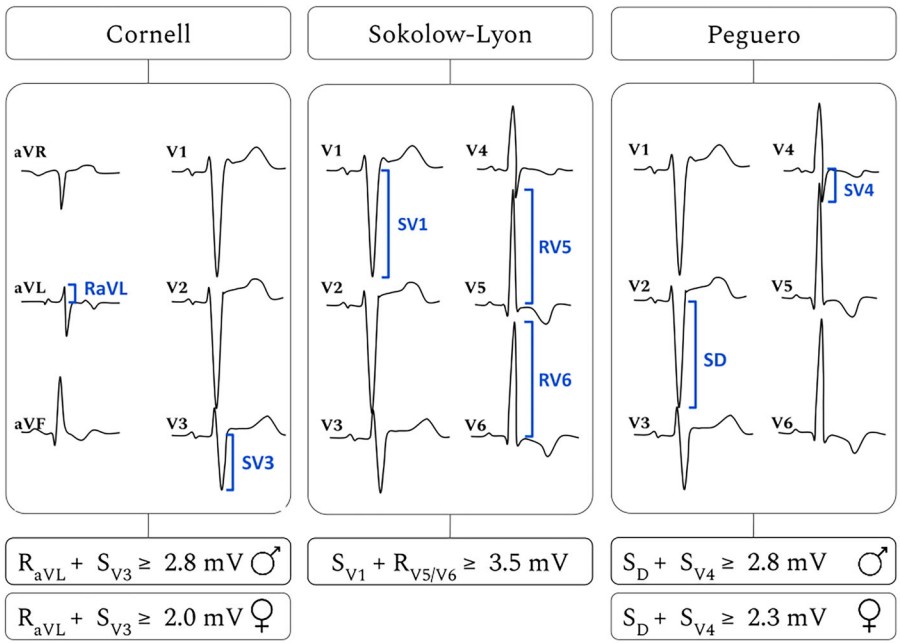

**Figure 1. Voltage-based ECG criteria for diagnosing LVH**
The Cornell and Peguero criteria are sex-specific, while the Sokolow–Lyon criterion does not account for sex.

the first column of Table 1, was exceeded. Ultimately, 30 models of cardiac hypertrophy were generated, 10 for each type of hypertrophy.

Using the volumetric mesh of the models, tissue volume and mass of the LV were computed for each scenario, assuming a myocardial density of 1.053 g/ml as referenced, e.g. in Vinnakota & Bassingthwaighte (2004). Following the methods outlined in the Methods section 'LVH ECG criteria', a height-based mass index was calculated. LVH was diagnosed when this mass index surpassed $51\,\mathrm{g\,m^{-2.7}}$.

Figure 2 displays the 40% cases of LVH from Table 1. Shown are eccentric and concentric cases in long-axis view in comparison to the healthy control model in the Methods section 'Anatomical model'. It is important to note that this figure focuses solely on the ventricles, the region of interest, while the kinematic growth algorithm was performed for the whole four-chamber geometry, excluding the torso.

## Activation sequences

The spatio-temporal transmembrane voltage data during the QRS phase, corresponding to ventricular depolarisation, were visualised for the control and hypertrophic models within the corresponding meshes.

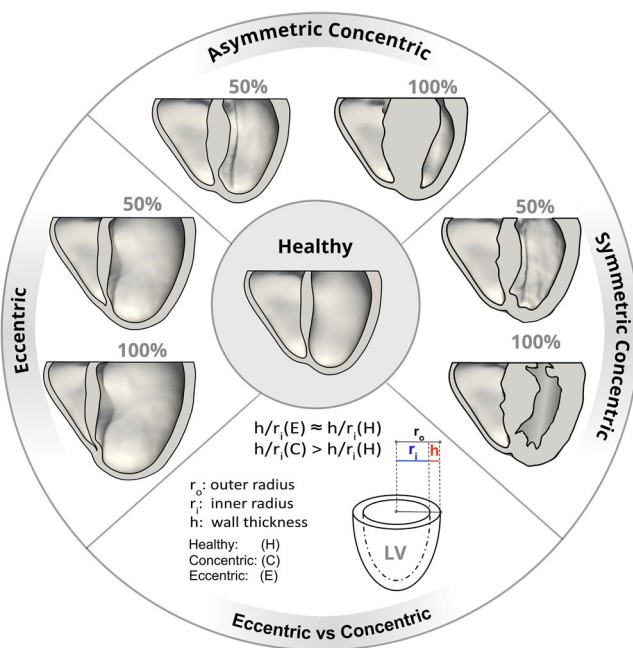

**Figure 2. Eccentric and concentric hypertrophy as predicted by serial and parallel sarcomere deposition models**
Eccentric hypertrophy resembles ventricular dilation. Symmetric concentric hypertrophy is characterised by uniform thickening of the left ventricular wall, in contrast to asymmetric concentric hypertrophy, which shows predominant thickening in the septal region. The lower part of the figure characterises eccentric *vs.* symmetric concentric hypertrophy.

The activation sequences were initiated on the endocardial surface in all cardiac models: three on the left sub-endocardial layer, identified as the LV anterior, posterior and septal fascicle, and two on the right sub-endocardial layer, identified as the moderator band and RV septal fascicle (see Methods sections 'Localisation of earliest activation sites' and 'Baseline ECGs' and Gillette, Gsell, Prassl, et al. (2021)). In the control model, four activation breakthrough sites were observed at the epicardial surface, indicated with white arrows in Fig. 3. These breakthroughs included one near the LV basal region, resulting from activation of the LV anterior fascicle; one near the apical region posteriorly, due to activation of the LV posterior fascicle; one at the anterior ventricular junction, resulting from septal activation of the ventricles; and one at the RV lateral wall, due to activation of the moderator band. The depolarisation waves stemming from the activation breakthroughs observed in the healthy model were contrasted with those in the hypertrophy models to highlight the differences in activation patterns resulting from the anatomical changes in the LV.

In the eccentric hypertrophy models, the site of breakthroughs on the epicardial surface were consistently similar across all models, closely resembling those observed in the healthy model. Additionally, the activation waves originating from the breakthroughs propagated nearly identically to those observed in the control, as depicted by the white arrows in the 50% and 100% eccentric cases in Fig. 3. In the concentric hypertrophy models, both in symmetric and asymmetric forms, a notable shift in the locations of epicardial breakthroughs was observed, with activation waves propagating distinctly differently compared to the control, as shown by the white arrows in the 50% and 100% concentric cases in Fig. 3. The symmetric concentric hypertrophy models with 10% to 100% wall thickening exhibited similar activation patterns to their asymmetric counterparts within the same range. In both cases, the global activation pattern was preserved, although epicardial activation was progressively delayed due to increased ventricular wall thickness. Beyond 50% hypertrophy, the symmetric concentric models maintained epicardial breakthrough at the anterior ventricular junction, in contrast to the asymmetric models, where this breakthrough site was lost. Furthermore, in the 50% to 100% range, the symmetric models showed a more pronounced activation delay in the LV posterior wall, particularly at the basal region, compared to the asymmetric models.

## Body surface potential maps

As ventricular dilation advanced, there was a noticeable increase in potential intensities on the torso's surface, especially around the precordial leads (Fig. 4, left). Higher

intensities near leads V2 and V3 at $T = 30$ ms and $T = 40$ ms, which correlated with higher R-wave and S-wave amplitudes in these ECG leads, were linked to larger activation waves originating from the expanded septal wall. The heightened potential near lead V4 at $T = 50$ ms in Fig. 4 was attributed to the activation of the LV apex in the endocardium, which was shifted closer to the torso due to dilation, along with its increased size, generating a larger electric field in the same region. Similarly, the increased intensities around leads V5 and V6 at $T = 50$ ms and $T = 60$ ms in Figs 4 and 5 were due to the expanded LV lateral wall also moving closer to the torso's surface. Note that $T = 0$ refers to the onset of ventricular depolarisation (the start of the QRS complex).

In asymmetric concentric LVH, non-uniform wall thickening disrupted the spatial coordination and timing of epicardial activation, leading to reduced extracellular potentials, particularly in regions corresponding to the lateral precordial leads at 50–60 ms in Figs 4 and 5. In contrast, symmetric concentric LVH with wall thickening exceeding 50% produced markedly stronger signals on BSPMs, most evident near leads V4 to V6. This was attributed to preserved epicardial activation synchrony and the presence of substantially thickened anterolateral and posterolateral walls, which intensified the extracellular potentials as the activation wavefront advanced toward the lateral aspect of the LV.

## Simulated ECGs

The QRS waves from standard 12-lead ECG signals were derived from the extracellular potential data computed in the simulations. The ECG leads were anatomically defined based on the positions of the physical leads used to record the measured ECG in the control model. The resulting signals from the hypertrophy models were compared to those from the control to identify morphological differences associated with the specific anatomical changes in the LV (see Fig. 6). In the eccentric hypertrophy models, the QRS complexes exhibited a progressive increase in R-wave and S-wave amplitudes in the precordial leads as ventricular dilation advanced, accompanied by a prolongation of QRS duration. In the V5 and V6 leads, the R-wave amplitude increased significantly, with peak values rising from 1.47 mV in lead V5 in the control model to 2.82 mV in the corresponding lead in the 100% eccentric growth case, representing an almost 92% increase in amplitude. In the V2 and V3 leads, the nadir of the S-wave also changed significantly, with the deepest value decreasing from −1.57 mV in lead V2 in the healthy model to −2.51 mV in the corresponding lead in the 100% eccentric growth case, reflecting an approximately 60% increase in amplitude. In advanced cases of eccentric hypertrophy, the observed R- and S-wave amplitudes in the precordial leads met the criteria

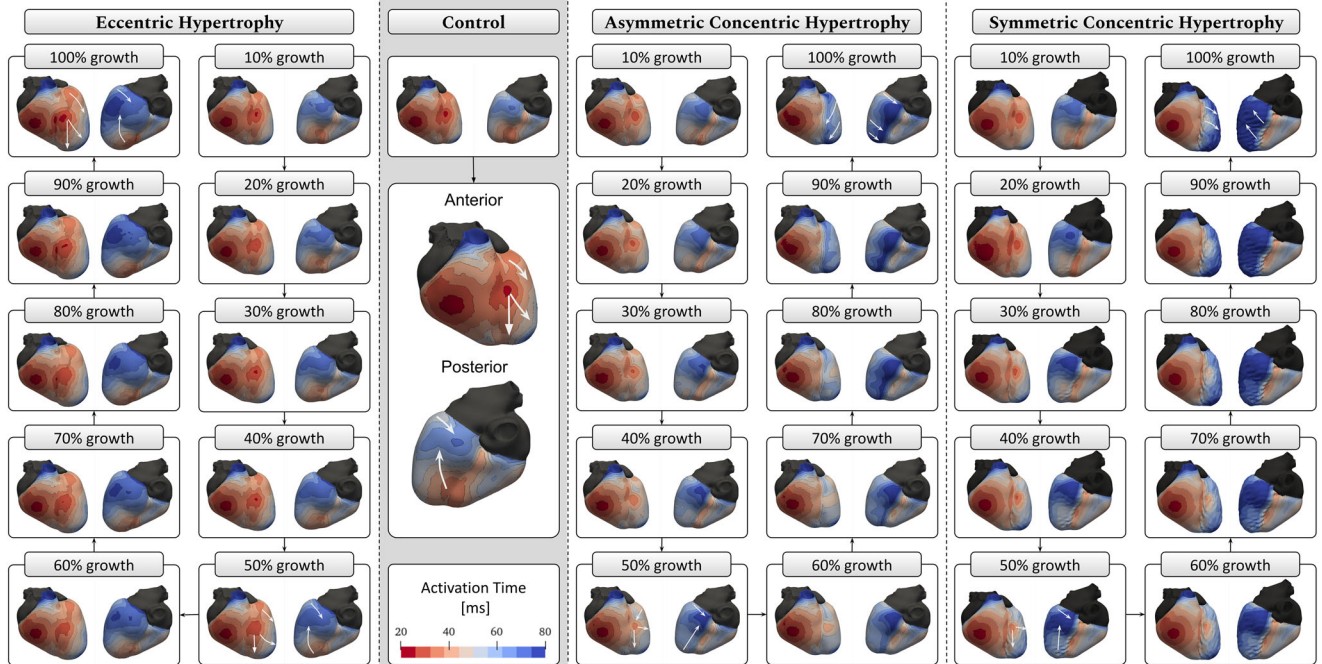

**Figure 3. Isochronal maps of ventricular depolarisation illustrating early activation sites and propagation patterns**
The eccentric hypertrophy model (first panel from the left) closely resembles the control (second panel), whereas concentric hypertrophy models (third and fourth panels) show shifts in early activation sites and altered wave propagation, as indicated by the white arrows, in the 50% and 100% cases.

set by the Sokolow–Lyon and Peguero–Lo Presti methods for diagnosing LVH.

In contrast to the eccentric models, QRS complexes in both asymmetric and symmetric concentric hypertrophy models did not exhibit a clear progression in amplitude with increasing growth levels. However, the symmetric models showed a more consistent and progressive pattern in QRS duration prolongation compared to their asymmetric counterparts, underscoring the effect of a uniform increase in wall thickness on global QRS duration.

R-wave fragmentation and voltage reduction were prominent features observed in the lateral limb leads in both concentric hypertrophy forms. Despite the morphological changes observed in ECGs from concentric hypertrophy models, they did not meet the established criteria for diagnosing LVH, as they did not exhibit sufficient amplitude increase in the relevant leads. Nonetheless, in asymmetric cases a notable increase in voltage was observed in lead V4, where the R-wave amplitude rose from 0.9 mV in the healthy model to 1.59 mV in the 100% concentric growth model,

representing an almost 75% increase. Also, the nadir of the S-wave in lead V1 decreased from −0.95 mV to −1.35 mV in the 100% wall-thickened model, reflecting a nearly 42% increase.

Similarly, in symmetric cases, the S-wave amplitude increased from −1.7 mV in the control model to −2.4 mV in the 100% growth model, reflecting a 40% increase.

Additionally, a voltage drop was noted in both forms of concentric LVH, particularly across leads II, III and aVF, and appeared more uniform in the symmetric cases.

## LVH ECG criteria

All simulated ECGs were systematically evaluated to ascertain whether the amplitude variations in the QRS complex conformed to established diagnostic criteria for LVH (see the Methods section 'LVH ECG criteria'. This analysis was conducted through an automated Python script algorithm, which integrated predefined LVH criteria, thereby ensuring a consistent and accurate assessment of the simulated ECGs.

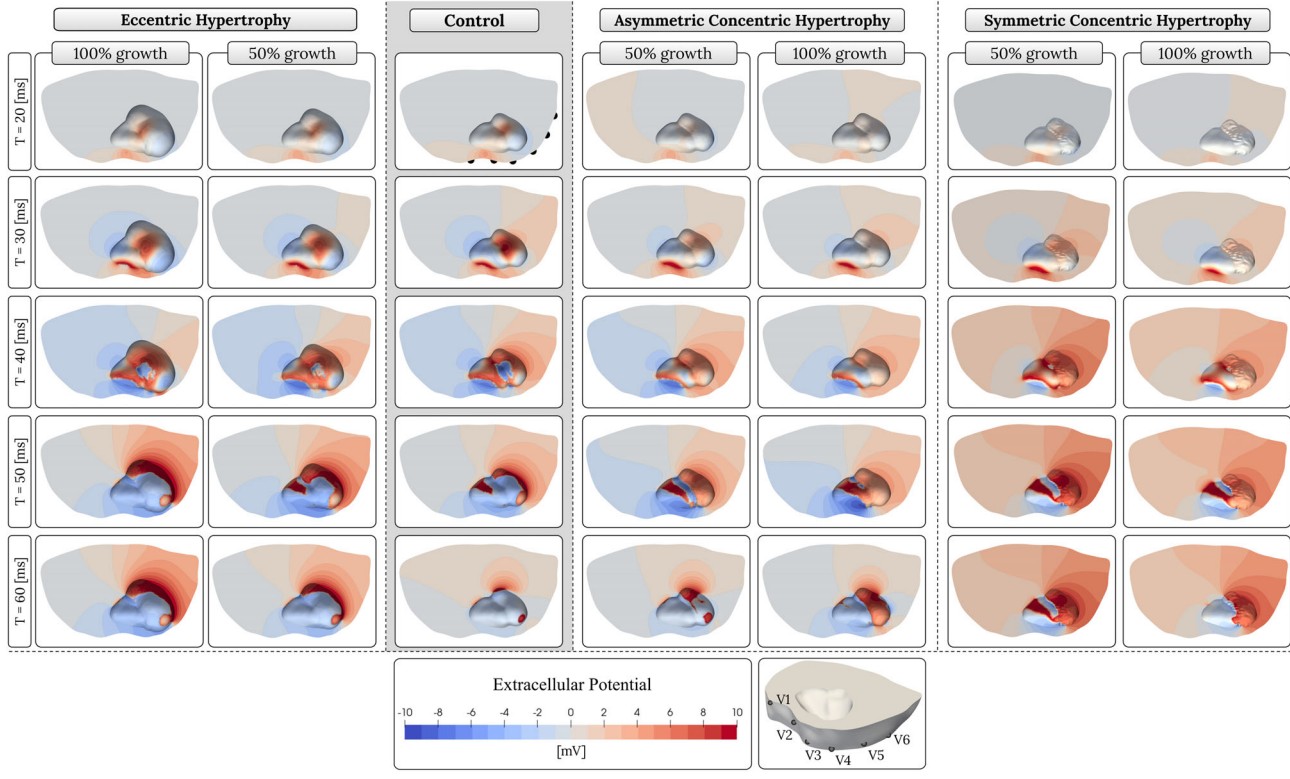

**Figure 4. Body surface potential maps illustrating the reflection of extracellular potentials during ventricular depolarisation, segmented transversely at the level of precordial leads across five time steps, with time 0 marking the onset of depolarisation**
The first panel from the left shows eccentric hypertrophy models, which exhibit a similar pattern of electrical potential propagation to the control (second panel), with increased intensity of potential reflection near precordial leads, particularly leads V2/V3 and V5/V6 at *T* = 50 ms. The third and fourth panels present concentric hypertrophy models, revealing a distinct pattern of electrical potential propagation compared to the control. Activation delay at the LV posterior site is evident in concentric models at *T* = 30 ms.

In the eccentric models with ventricular dilation ranging from 40% to 100%, which were classified as hypertrophy events based on the mass indexation method, all cases satisfied the Sokolow–Lyon criterion (7 out of 7 cases). Among these, the models with 60–100% ventricular dilation met the Peguero–Lo Presti criterion in 5 out of 7 cases, while only the model with 100% dilation fulfilled the Cornell criterion (1 out of 7 cases), see Table 2.

In asymmetric concentric models, none of the hypertrophy cases – characterised by more prominent septal wall thickening – met any of the established LVH diagnostic criteria, see Table 3.

Unlike the outcomes observed with eccentric hypertrophy, the sum of the S-wave amplitude in lead V1 and the R-wave amplitude in either lead V5 or V6 did not exceed 3.5 mV in any asymmetric concentric cases, thereby failing to satisfy the Sokolow–Lyon criterion. Furthermore, the sum of the deepest S-wave amplitude in the precordial leads (V2 in the simulated ECGs) and the S-wave amplitude in lead V4 did not reach the minimum threshold of 2.8 mV, thus not meeting the Peguero–Lo Presti criterion. Similarly, the R-wave in lead aVL and the S-wave in lead V3 did not reach the 2.8 mV threshold required to meet the Cornell voltage criterion.

In symmetric cases, 3 out of 7 met the Sokolow criterion, although none satisfied the Cornell or Peguero criteria, similar to the asymmetric cases, see Table 4. However, it is worth noting that the values for non-detected cases in symmetric models were very close to the Sokolow criterion LVH detection threshold.

## ECG characteristics of LVH in clinical data

For the independent analysis of the clinically recorded ECGs (Methods section 'Analysis of clinical ECGs') the comparison of maximum QRS amplitudes revealed that,

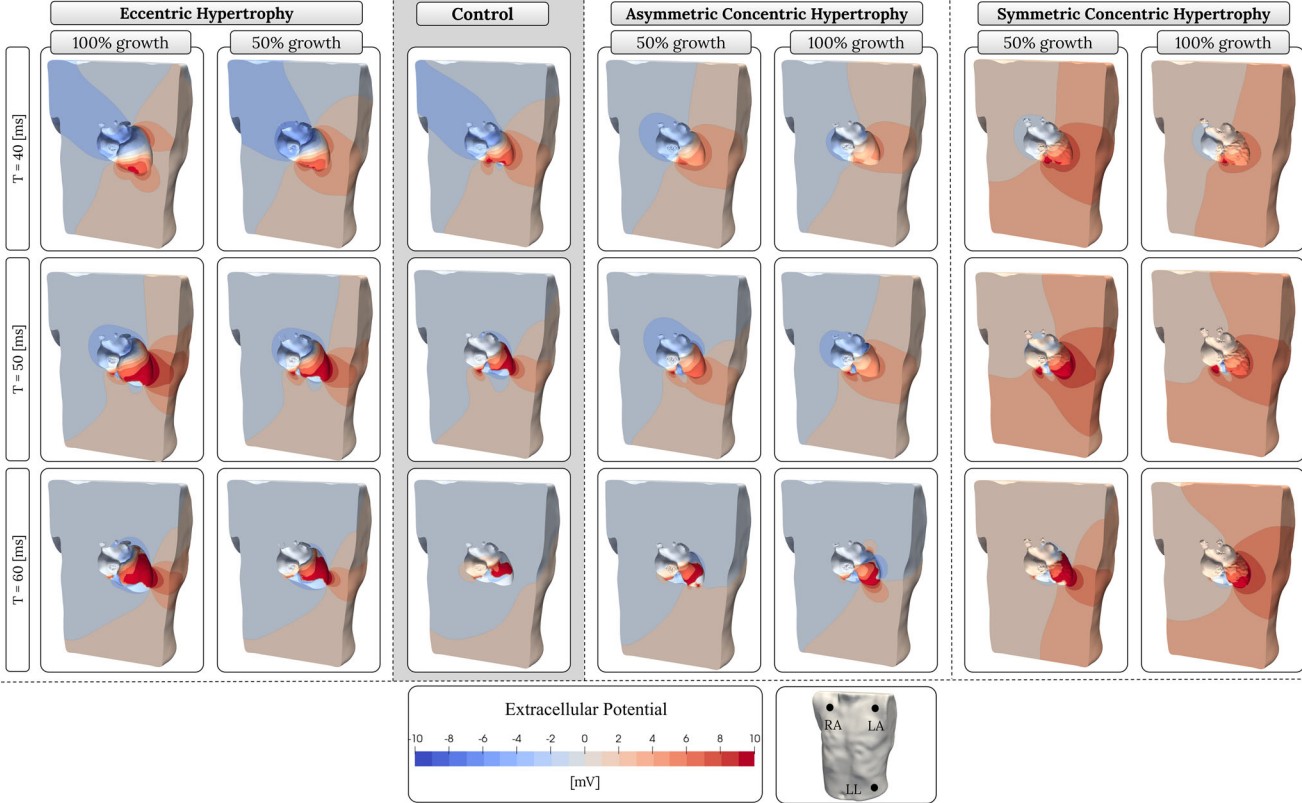

**Figure 5. Body surface potential maps showing the distribution of extracellular potentials on the posterior torso, segmented in the coronal plane across three time points, with time zero marking the onset of ventricular depolarisation**

The leftmost panel presents the eccentric hypertrophy models, which exhibit a pattern comparable to the control (second panel), but with greater potential intensity on the left lateral torso, particularly near leads V5 and V6 at $T$ = 50 ms. The third and fourth panels illustrate the asymmetric and symmetric concentric hypertrophy models, respectively, both exhibiting distinct extracellular potential propagation patterns compared to the control. Symmetric models also exhibit stronger extracellular potentials on the left lateral torso, attributable to the increased thickness of the anterolateral and posterolateral walls.

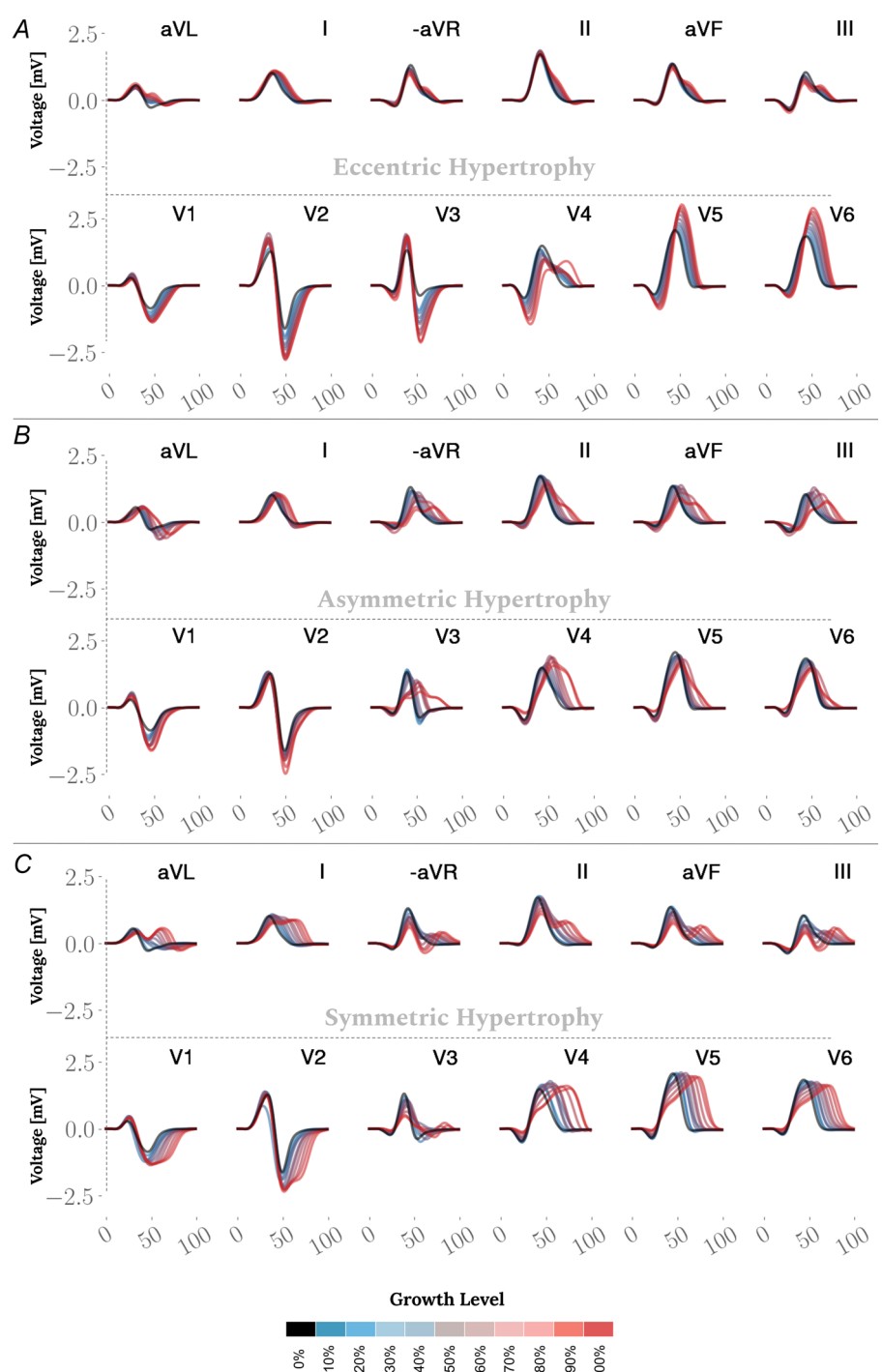

**Figure 6. Comparison of QRS complex patterns in eccentric *vs.* concentric hypertrophy models from 12-lead ECGs**

*A*, ECGs from eccentric hypertrophy models show a progressive increase in R- and S-wave amplitudes in the precordial leads, accompanied by prolonged QRS duration. *B*, ECGs from asymmetric concentric hypertrophy cases exhibit prolonged QRS duration without significant changes in voltage. *C*, in symmetric hypertrophy models, a clearer progression in QRS duration prolongation is observed compared to asymmetric counterparts. Similar to asymmetric LVH cases, they do not exhibit a progressive increase in QRS amplitude.

**Table 2. Results of ECG analysis based on established LVH criteria for both healthy and eccentric cardiac hypertrophy models**

| Cardiac Models | | Sokolow | Cornell | Peguero |
|---|---|---|---|---|
| | | 3.5mV | 2.8mV | 2.8mV |
| Healthy | | 2.91 | 1.31 | 1.93 |
| Eccentric 10% | | 3.16 | 1.61 | 2.22 |
| Eccentric 20% | | 3.37 | 1.68 | 2.28 |
| Eccentric 30% | | **3.57** | 1.82 | 2.47 |
| Eccentric 40% | ★ | **3.74** | 1.97 | 2.57 |
| Eccentric 50% | ★ | **3.95** | 2.01 | 2.60 |
| Eccentric 60% | ★ | **4.27** | 2.18 | **2.86** |
| Eccentric 70% | ★ | **4.38** | 2.42 | **2.92** |
| Eccentric 80% | ★ | **4.57** | 2.46 | **3.05** |
| Eccentric 90% | ★ | **4.74** | 2.67 | **3.08** |
| Eccentric 100% | ★ | **4.87** | **2.85** | **3.31** |
| Sensitivity | | 100% | 14% | 71% |
| Specificity | | 75% | 100% | 100% |
| Precision | | 100% | 100% | 100% |
| F-Score | | 86% | 24.56% | 84.04% |

All seven actual hypertrophy cases met the Sokolow-Lyon criterion, five of which also met the Peguero–Lo Presti criterion, while only one case met the Cornell voltage criterion. Values exceeding the corresponding criterion thresholds are highlighted in bold. Hypertrophy models identified by the mass indexation method are marked with ★.

**Table 3. ECG analysis based on established LVH criteria for asymmetric concentric hypertrophy models**

| Cardiac Models | | Sokolow | Cornell | Peguero |
|---|---|---|---|---|
| | | 3.5mV | 2.8mV | 2.8mV |
| Healthy | | 2.91 | 1.31 | 1.93 |
| Asym. Concentric 10% | | 3.03 | 1.40 | 2.05 |
| Asym. Concentric 20% | | 3.05 | 1.33 | 2.18 |
| Asym. Concentric 30% | | 3.14 | 1.35 | 2.19 |
| Asym. Concentric 40% | ★ | 3.27 | 1.31 | 2.13 |
| Asym. Concentric 50% | ★ | 3.39 | 1.22 | 2.11 |
| Asym. Concentric 60% | ★ | 3.26 | 0.64 | 2.06 |
| Asym. Concentric 70% | ★ | 3.17 | 0.63 | 2.31 |
| Asym. Concentric 80% | ★ | 3.16 | 0.75 | 2.18 |
| Asym. Concentric 90% | ★ | 3.15 | 0.49 | 2.30 |
| Asym. Concentric 100% | ★ | 3.09 | 0.45 | 2.49 |
| Sensitivity | | 0% | 0% | 0% |
| Specificity | | 100% | 100% | 100% |
| Precision | | N/A | N/A | N/A |
| F-Score | | N/A | N/A | N/A |

The resulting changes in amplitude in the ECGs from the asymmetric concentric hypertrophy models did not lead to meeting any of the established criteria. Hypertrophy models identified by the mass indexation method are marked with ★.

on average, leads aVL, I, III, V1, V5 and V6 in the LVH cohort exhibited at least a 50% higher amplitude than their counterparts in the normal cohort, indicating a high sensitivity to LVH. In the LVH cohort, Lead aVL demonstrated the highest differential in maximum amplitude at 93.7%, followed by Lead III at 77.3%, and Lead V1 at 69%, when compared to their counterparts in the normal cohort.

## Discussion

### Activation sequences

Activation sequences in the heart provide a valuable framework for assessing cardiac function. Comparing these patterns between a healthy control model and test cases reveals anomalies in diseased models, enhancing our understanding of both pathological mechanisms and

**Table 4. ECG analysis based on established LVH criteria for symmetric concentric hypertrophy models**

| Cardiac Models | | Sokolow | Cornell | Peguero |
|---|---|---|---|---|
| | | 3.5mV | 2.8mV | 2.8mV |
| Healthy | | 2.91 | 1.31 | 1.93 |
| Sym. Concentric 10% | | 3.2 | 0.76 | 1.79 |
| Sym. Concentric 20% | | 3.49 | 1.04 | 2.13 |
| Sym. Concentric 30% | | 3.36 | 0.78 | 2.02 |
| Sym. Concentric 40% | ★ | **3.51** | 0.69 | 2.32 |
| Sym. Concentric 50% | ★ | 3.42 | 0.92 | 2.43 |
| Sym. Concentric 60% | ★ | **3.63** | 0.69 | 2.35 |
| Sym. Concentric 70% | ★ | **3.5** | 0.72 | 2.46 |
| Sym. Concentric 80% | ★ | 3.46 | 0.71 | 2.46 |
| Sym. Concentric 90% | ★ | 3.43 | 0.64 | 2.44 |
| Sym. Concentric 100% | ★ | 3.43 | 0.92 | 2.52 |
| Sensitivity | | 37.5% | 0% | 0% |
| Specificity | | 100% | 100% | 100% |
| Precision | | 54.5% | N/A | N/A |
| F-Score | | 54.5% | N/A | N/A |

Three out of seven met the Sokolow criterion, while none satisfied the Cornell or Peguero criteria. Hypertrophy models identified by the mass indexation method are marked with ★.

normal physiology to improve the identification of cardiac abnormalities.

The models exhibiting ventricular dilation (eccentric LVH) showed significant similarity to the control model. Despite the considerable increase in LV size, the activation patterns in the eccentric models consistently mirrored those of the control. A slight temporal delay was observed that can be attributed to the larger LV requiring additional time for activation to spread across the LV. In contrast, the concentric models showed noticeable deviations in the spatial distribution of activation across the epicardial surface compared to the control model. The sites of earliest epicardial activation in the LV differed significantly from those in the healthy control model, with differences becoming more pronounced when wall thickness exceeded a 50% threshold.

More specifically, in the 10% to 50% hypertrophy range, symmetric concentric models exhibited activation behaviour comparable to that of asymmetric concentric models. However, beyond 50% hypertrophy, a divergence emerged: asymmetric models lost the epicardial breakthrough at the anterior ventricular junction, whereas symmetric models retained it. This difference was attributed to the markedly thickened anteroseptal LV wall in the asymmetric models, which impeded transmural propagation of the activation wave. In contrast, the anteroseptal wall in the symmetric models was comparatively less thickened, allowing preservation of breakthrough at this anatomical site, albeit with some delay. Additionally, symmetric models with hypertrophy beyond 50% exhibited a notable delay in activation of the posterior LV wall, particularly in the basal region. As

observed in the control model, epicardial activation of the basal posterior region was typically achieved via wavefronts originating from both the basal anterior and apical posterior regions. In the hypertrophied models, both pathways were delayed due to increased wall thickness, resulting in a pronounced delay at the basal posterior epicardium.

In general, although EP remodelling was not incorporated in the simulations, increased wall thickness alone substantially altered the distribution of electrical potentials, resulting in delayed epicardial activation and, consequently, a modified epicardial activation pattern.

## Body surface potential maps

BSPMs provide a detailed spatial representation of the heart's electrical activity across the torso's surface. Unlike standard ECGs, which are limited by a restricted number of leads, BSPMs offer a broader view of the heart's electrical dynamics, providing enhanced spatial information that deepens our understanding of the reflection of activation waves on the body surface. Therefore, BSPMs can potentially offer valuable insights into the underlying mechanisms of complex cardiovascular pathologies, which we leveraged in this study to interpret the ECGs derived from the hypertrophy models.

The observed consistency in the BSPM patterns across different eccentric cardiac models (Figs 4 and 5, left) goes hand in hand with results discussed above that ventricular dilation does not significantly alter the default activation pattern in the LV. In contrast to that, in cases of concentric hypertrophy the BSPM patterns became

less consistent and deviated from those observed in the control model (Figs 4 and 5, right). This was especially pronounced when wall thickness increased by more than 50%, resulting in a distinct surface potential map on the torso that aligned with the changed activation patterns seen in the corresponding cardiac models. The differential impact of eccentric and concentric hypertrophy highlights a question: why did the progressive increase in ventricular wall thickness observed in concentric hypertrophy not produce corresponding changes in these electrocardiographic parameters, as seen in eccentric hypertrophy? This discrepancy can be attributed to the distinct cellular and structural growth mechanisms underlying each type of hypertrophy. In both symmetric and asymmetric concentric hypertrophy forms, sarcomere deposition occurs in a parallel fashion, resulting in a greater increase in wall thickness; therefore, LV growth is more pronounced in the transmural direction than in the apicobasal direction.

Given that the activation wave had to travel from the endocardial to the epicardial surface, both symmetric and asymmetric increases in wall thickness resulted in prolonged transmural propagation time. This delay led to discordant epicardial breakthroughs and activations, ultimately producing more distinct QRS peak amplitudes and morphologies – a feature that was absent in eccentric hypertrophy.

Symmetric hypertrophy cases, as shown in Figs 4 and 5, exhibited stronger surface potentials, particularly near the lateral leads, due to the larger electric field generated by the thickened lateral wall. However, this was not observed in the ECGs, suggesting the need for further investigation. A deeper analysis revealed that the ECG lead placement was suboptimal for capturing the increased amplitude in this model. Specifically, the precordial leads were misaligned along the horizontal plane, and the cut plane was positioned above the lateral leads (V3–V6), resulting in an intensity mismatch between the figure and the actual ECG recordings. These results highlighted that the detection of LVH was influenced by the precise positioning of ECG leads. Slight misalignment in lead placement was found to affect the accurate recording of surface potentials, which in turn could impact the efficacy of LVH detection through surface ECGs.

At $T = 40$ ms, i.e. 40 ms in Fig. 5, The negative potential directed toward the right arm, corresponding to lead aVR, was more pronounced in the control and eccentric hypertrophy models, whereas both symmetric and asymmetric concentric models showed a clear reduction in intensity.

This negative potential is attributed to LV endocardial activation, which renders the blood volume within the LV cavity negative, thereby reflecting a negative potential toward the right arm. Due to the markedly reduced blood volume within the LV in both forms of concentric hyper-

trophy, the negative potential directed toward the right arm was substantially attenuated.

In eccentric models at $T = 50$ ms in Fig. 5, higher extracellular potentials were generated in the torus model due to the expanded activated region of the LV, whereas in concentric models, the lower potentials reflected activation delays resulting from increased wall thickness. The same finding was also visible at $T = 50$ ms in Fig. 4.

Another relevant observation is that both forms of hypertrophy affected the RV activation pattern. In particular, under the 100% eccentric LVH condition, RV activation was noticeably accelerated. This was clearly evident at 50 ms, when the RV region within the torso conductor volume exhibited a completely negative potential, indicating that the RV had already undergone depolarisation. This earlier RV activation in eccentric LVH cases was due to larger activation waves reaching the RV from the LV side, arriving from both the anterior and posterior directions. In contrast to eccentric LVH, both symmetric and asymmetric forms of concentric LVH resulted in delayed RV activation. This delay was observed at both 50% and 100% growth levels, with a more pronounced effect at the 100% level. The delay was attributed to the extended travel time of LV endocardial activation towards the RV, as LV activation also plays a role in RV activation.

## Simulated ECGs

The ECG has been a key instrument in diagnosing LVH over the past decades due to its affordability, ease of set-up and quick usability. It provides real-time results in a non-invasive manner, offering numerous advantages that make it a practical and favourable choice for both patients and clinicians.

In the context of hypertrophic cardiomyopathy, LVH diagnosis from 12-lead ECG signals has been a controversial topic for decades (Marcato et al., 2022). Historically, the absence of effective computer-aided tools, and even now with the advent of ML in healthcare, debates about its efficacy persist (Krittayaphong et al., 2013; Rabkin, 2024). As this study demonstrated, the cardiac function in terms of potential distribution is inherently three-dimensional. Thus, recording local potentials on the surface of the torso with a limited number of electrodes significantly limits the input data necessary to understand such complex cardiac events. Nevertheless, given the numerous advantages of ECG in clinical practice, enhancing its application by improving diagnostic criteria is indeed worthwhile. Therefore, our aim was to offer new insights into LVH from a different perspective, simplifying the event and its mechanism that alter ECG morphology, supported by computational simulations.

ECGs from hypertrophic models showed distinct changes depending on the specific anatomical alterations in the LV. In cases of ventricular dilation or eccentric hypertrophy, the QRS segment typically exhibited increased R- and S-wave amplitudes in the precordial leads, particularly V2/V3 and V5/V6, consistent with findings from clinical data in the PhysioNet data-set, which identified these leads as sensitive indicators of hypertrophy (Fig. 7). Unlike eccentric hypertrophy, which was characterised solely by increased amplitudes, both forms of concentric hypertrophy exhibited a combination of increased and decreased amplitudes across different leads, reflecting more complex underlying mechanisms. Both symmetric and asymmetric concentric hypertrophy led to a notable increase in S-wave amplitude in leads V1 and V2, with V1 also being among the most sensitive leads in ECGs with LVH according to clinical data from the PhysioNet dataset (Fig. 7). All concentric hypertrophy models exhibited no increase in R-wave amplitude in leads V5 and V6, a pattern that contrasts distinctly with the amplified R-waves observed in eccentric hypertrophy. An important observation is that eccentric hypertrophy primarily affected the precordial leads, with minimal changes in the limb leads, whereas both forms of concentric hypertrophy induced alterations across all 12 leads.

In evaluating QRS duration, distinct patterns of prolongation were observed between eccentric and concentric hypertrophy models (see Fig. 8). Specifically, eccentric hypertrophy models exhibited a more pronounced QRS prolongation in the precordial leads, while asymmetric concentric hypertrophy models showed greater prolongation in the limb leads. Notably, symmetric concentric hypertrophy models resulted in prolonged QRS duration across all 12 leads. Another important observation is that eccentric hypertrophy and symmetric concentric hypertrophy displayed a clear progression in QRS duration in line with the magnitude of growth, whereas asymmetric concentric hypertrophy exhibited a less pronounced progression (see Fig. 8). In the control model, QRS duration was 52 ms, increasing to 65 ms (+13 ms) in both 100% eccentric and 100% asymmetric concentric hypertrophy models. Symmetric concentric hypertrophy, however, led to an even more pronounced prolongation, reaching 73 ms (+21 ms). This corresponds to 25% and 40% increases in LV depolarisation, respectively.

It is important to note that the observed increase in QRS duration during ventricular depolarisation was solely attributed to anatomical growth, without any concurrent EP remodelling effects.

It can be concluded that in cases of eccentric hypertrophy, where pathological progression has not yet caused severe electrical remodelling or significant structural changes like fibrosis – both of which can alter conduction velocity and modify the activation sequence in the LV – an increase in voltage detected by precordial leads is expected. This increase is primarily due to the expansion of the LV, which brings the apex and lateral wall closer to the torso surface. This proximity allows the leads to record higher voltages. Additionally, the increase in myocardial mass generates a larger electrical field, leading to higher amplitudes on the ECG. These factors likely contribute to the characteristic ECG changes observed in eccentric hypertrophy before extensive pathological remodelling takes place.

In concentric hypertrophy, even under optimal conditions where anatomical changes had not yet affected EP adaptation or structural fibre alterations, the significant wall thickening created a distinct environment for the propagation of activation sequences. Uniform ventricular wall growth, or symmetric hypertrophy, may maintain a normal ECG morphology, regardless of QRS duration. In contrast, asymmetric concentric hypertrophy, due to non-uniform transmural wall thickening, is more likely to result in noticeable alterations in ECG morphology.

ECG findings in both forms of concentric hypertrophy revealed that, despite increased myocardial mass and wall thickness, certain leads, such as V5 and V6, showed a reduction in voltage. These observations align with those reported in studies that identified a decrease in maximum QRS amplitude in LVH models (Bacharova, 2007; Bacharova et al., 2004, 2005).

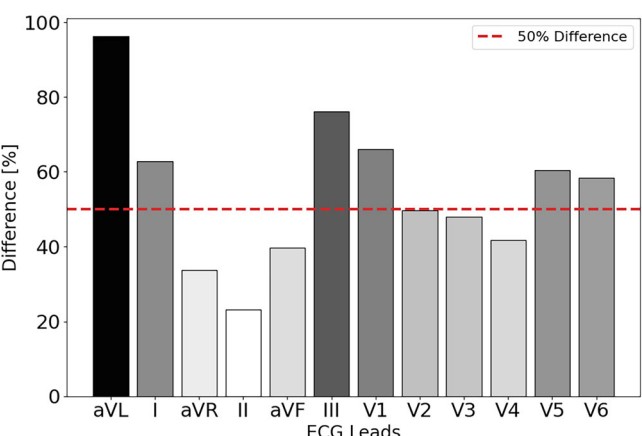

**Figure 7. Leads sensitivity analysis for LVH in the PTB-XL ECG data-set from PhysioNet, encompassing 3937 records: 984 normal male ECGs, 983 normal female ECGs, 989 female ECGs with LVH, and 981 male ECGs with LVH**
The average maximum amplitude across all leads of 12-lead ECGs for both LVH and normal cohorts is calculated with a 95% truncated mean. The percentage difference in amplitude between the normal and LVH cohorts (eqn (3)), is highlighted, showing that leads aVL, III and V1 were the most sensitive to LVH, with significantly increased amplitude in the LVH cohort compared to the normal cohort. Colour-coded map where black represents the highest sensitivity, gradually decreasing to lighter shades indicating lower sensitivity.

This occurred because the thickened walls caused delays in epicardial activation and disrupted synchrony among epicardial breakthroughs. As a result, local activation waves spread independently rather than merging into a coherent activation wave. This effect was noticeable even under our assumed conditions without EP remodelling. However, if fibrosis or other pathological changes were present, the discrepancy between expected and observed voltage changes in the ECG would likely be even more pronounced (see also Appendix D).

### ECG based LVH diagnosis

Over the past few decades, more than 30 ECG-based criteria have been introduced for the diagnosis of left ventricular hypertrophy (LVH) (Hancock et al., 2009). These criteria primarily rely on the amplitude of R- and S-waves in leads reflecting left ventricular activity. However, factors such as sex, ethnicity, age, obesity and physical fitness can influence cardiac dimensions and modulate the surface-detected electrical signals. This inter-individual variability reduces the sensitivity of ECG-based LVH detection and poses challenges for consistent diagnosis across diverse populations. Among the most widely used ECG criteria for identifying LVH are the Sokolow–Lyon and Cornell criteria. The Sokolow–Lyon approach assesses voltages in leads V1 and V5/V6, while the Cornell criterion is based on measurements from leads aVL and V3 (see Fig. 1). Building on this, our analysis of the clinical dataset revealed that leads aVL, III and V1 were particularly sensitive for detecting LVH, as determined through lead sensitivity analysis using the PTB-XL ECG dataset (see the Results section 'ECG characteristics of LVH in clinical data').

Although the leads recommended by the aforementioned criteria overlap with those identified as sensitive in our clinical ECG-based sensitivity analysis, the use of fixed voltage thresholds in these criteria is widely recognised as a limitation, often resulting in suboptimal sensitivity. Tailoring these thresholds to the characteristics of the target population may enhance diagnostic accuracy by accounting for inter-individual and group-level variability in LVH expression. Moreover, our computational simulations demonstrated that anatomical changes, particularly LV dilation and wall thickening, exert distinct effects on ECG morphology. This variability challenges the development of a universal criterion capable of reliably detecting all forms of LVH. Leads aVL and V1 were found to be more sensitive to concentric hypertrophy, while leads V2/V3 and V5/V6 demonstrated greater responsiveness to eccentric hypertrophy (Fig. 6). Given that concentric LVH carries a higher clinical risk than eccentric LVH (Guzik et al., 2021), the presented findings may support the development of ECG criteria tailored to the specific characteristics of each LVH subtype.

Through simulation results and insights drawn from the discussions above, it was determined that leads aVL, V1 and V5/V6 were the most sensitive to anatomical changes. Independent sensitivity analyses of ECG leads in the PTB-XL dataset confirmed these findings, with leads aVL, III, V1, V5 and V6 being identified as particularly sensitive to LVH (Fig. 7). Incorporating these leads into a revised ECG criterion, alongside population-specific threshold adjustments, may improve sensitivity and specificity compared to existing criteria.

Finally, while ML has significantly advanced the diagnosis of cardiac events through ECG analysis, ML tools for diagnosing LVH from ECGs have fallen short to outperform specificity of classic ECG criteria (Rabkin, 2024). Moreover, the reliance of ML tools on model architecture, data quality and quantity, or training techniques often limit their interpretability to a 'black box' approach, where clinicians may find it challenging to understand the decision-making process (Kelly et al., 2019; Scott et al., 2024). This lack of transparency stands in stark contrast to conventional methods that

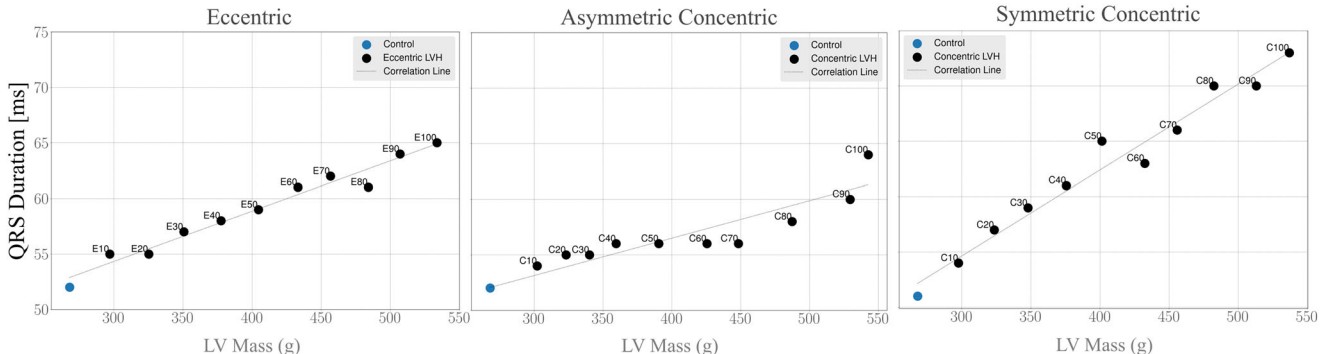

**Figure 8. QRS duration prolongation in left ventricular hypertrophy**
A positive association was observed between ventricular mass and QRS duration across all hypertrophy types, with the relationship being more pronounced in eccentric and symmetric concentric hypertrophy.

provide straightforward, human-readable frameworks for diagnosing conditions such as LVH from ECGs. Consequently, these transparent methodologies may remain preferable in clinical settings, as they facilitate more intuitive and direct application by healthcare professionals.

### Limitations and future directions

First, in a real-world context, the progression of hypertrophy encompasses not only anatomical growth but also extensive structural and EP remodelling, rendering it a highly complex cardiac condition. Structural remodelling includes alterations in fibre orientation and increased fibrogenesis, while EP remodelling affects electrical conductivity and CV (Bacharova et al., 2010; Tseng et al., 2006; Weber & Brilla, 1991; Wu et al., 2017; see also Appendix D for more details). Initially, these remodelling processes function as compensatory mechanisms in response to pressure or volume overload. However, when this adaptation persists over time, it transitions into a maladaptive phase (Schiattarella & Hill, 2015). For instance, prolonged hypertrophy-driven fibrogenesis modifies the heterogeneity of myocardial conductivity, ultimately influencing CV. This change, combined with the presence of fibrosis, can establish a foundation for reentry circuits, significantly heightening the risk of arrhythmias (Bacharova et al., 2023; Chatterjee et al., 2014). Consequently, the risks associated with hypertrophy are not only significant but also time-dependent, emphasising the importance of timely medical intervention to prevent its progression and avert potentially life-threatening cardiac outcomes. While hypertrophy mechanisms are undeniably complex, this study focused specifically on anatomical growth to simplify the analysis and improve understanding, isolating it from EP remodelling effects in LVH. Conductivities and conduction velocities from the control model were consistently preserved across all hypertrophic models, and anatomical locations of the primary fascicles remained unchanged in UVCs. We believe the insights gained from this approach provide a novel perspective on the complex nature of hypertrophy. Extending this work to include EP remodelling offers an exciting direction for future research.

Second, this study was limited by the availability of healthy, high-fidelity models personalised to an ECG, restricting us to a single physiologically detailed cardiac mesh as a control model, alongside the generation of 30 hypertrophy models derived from this reference. Although the cohort size is small, it is crucial to emphasise that the results align well with clinical data and do not compromise the key conclusions of this work, ensuring the validity and relevance of our findings. Also variations in longitudinal and transverse fibre orientations would have

an effect on the signal, but they would also influence the fitting procedure to the clinically measured ECG signal, which is not the main focus of this paper. Nevertheless, expanding the cohort size in future studies and assessing hypertrophy mechanisms across a broader range of virtual heart models would likely enhance the robustness and broader applicability of our results.

Third, access to clinical data remains a significant challenge. Comparative studies between non-LVH and LVH ECGs across different data-sets could offer additional insights into the distinguishing features of these conditions. The availability of the Physionet Dataset was instrumental in facilitating our ECG analysis in this study (for which we express our deep gratitude). Future research that incorporates a wider range of clinical data-sets could deepen our understanding of LVH and its electrophysiological manifestations. This approach may also lead to the development of new diagnostic criteria – based on considerations above – that can be thoroughly tested and compared against established criteria using a larger, more diverse cohort.

### Conclusion

The computational simulations confirmed clinical knowledge that eccentric and concentric LVH distinctly affect 12-lead ECG signals. Eccentric hypertrophy primarily impacted the precordial leads, exhibiting significant amplitude increases, particularly in leads V2/V3 and V5/V6. In contrast, concentric hypertrophy affected all leads, with more pronounced changes observed in the limb leads (aVL and III) and specific precordial leads (V1 and V4), demonstrating a mix of amplitude increases and decreases. Both eccentric and concentric LVH consistently prolonged the QRS complex by up to 21 ms – a 40% increase from baseline – despite the absence of EP remodelling factors typically associated with LVH. Furthermore, eccentric hypertrophy accelerated RV depolarisation due to septal expansion, whereas concentric hypertrophy caused delayed RV activation linked to the thickened inter-ventricular septum. Leads aVL, III, V1 and V5/V6 were identified as the most sensitive to LVH, with computational results aligning well with independent clinical data.

By increasing diagnostic accuracy, our findings could further strengthen the role of established ECG criteria as valuable, cost-effective additions to clinical assessments and imaging, supporting better detection and management of LVH.

### Appendix A: Tissue mechanics

Mechanical properties of cardiac tissue were described as a hyperelastic, nearly incompressible, and anisotropic material with a non-linear stress–strain relationship. The

deformation gradient $\mathbf{F}$ characterises the deformation $\mathbf{u}$ of the tissue, mapping it from its reference configuration $\Omega_0(\mathbf{X})$ to its current configuration $\Omega_t(\mathbf{x})$:

$$F_{ij} = \frac{\partial x_i}{\partial X_j}, \quad i, j = 1, 2, 3. \tag{4}$$

By convention, we define $J = \det\mathbf{F} > 0$, and introduce the right Cauchy–Green tensor $\mathbf{C} = \mathbf{F}^\top\mathbf{F}$. To model the nearly incompressible behaviour or the elastic material, we used a multiplicative decomposition of the deformation gradient, following the approach by Flory (1961):

$$\mathbf{F} = J^{1/3}\,\bar{\mathbf{F}}, \mathbf{C} = J^{2/3}\,\bar{\mathbf{C}}, \quad \text{with} \quad \det\bar{\mathbf{F}} = \det\bar{\mathbf{C}} = 1. \tag{5}$$

In this work, we described mechanical deformation by the quasi-static equilibrium equations:

$$-\mathrm{Div}\left[\mathbf{F}\mathbf{S}\left(\mathbf{u}, \mathbf{X}\right)\right] = \mathbf{0} \quad \text{for } \mathbf{X} \in \Omega_0 \times (0, T), \tag{6}$$

where $\mathbf{S}(\mathbf{u},\mathbf{X})$ is the second Piola–Kirchhoff stress tensor and Div denotes the divergence operator in the reference configuration.

Passive stresses were modelled based on the constitutive equation

$$S_{\mathrm{p}} = 2\frac{\partial\Psi\left(\mathbf{C}\right)}{\partial C},$$

where $\Psi$ is a strain-energy function to model the anisotropic behaviour of cardiac tissue.

To model the mechanical properties of all four chambers, we adopted a reduced Holzapfel–Ogden material with fibre dispersion (Gültekin et al., 2016):

$$\Psi\left(\mathbf{C}\right) = \frac{\kappa}{2}\,\ln\left(J\right)^2 + \frac{a}{2b}\exp\left[b\left(\bar{I}_1 - 3\right)\right]$$
$$+ \frac{a_{\mathrm{f}}}{2b_{\mathrm{f}}}\left\{\exp\left[b_{\mathrm{f}}\left(\bar{I}_{\mathrm{f}}^* - 1\right)^2\right] - 1\right\} \tag{8}$$

with $\mathbf{f}_0$ the primary orientation of myocytes; $\kappa > 0$ being a penalty parameter for enforcing the nearly incompressible behaviour of the tissue; stress-like parameters $a > 0$, $a_{\mathrm{f}} > 0$ and dimensionless material parameters $b > 0, b_{\mathrm{f}} > 0$; and invariants:

$$\bar{I}_1 = tr\bar{\mathbf{C}}, \quad \bar{I}_{\mathrm{f}*} = \kappa_{\mathrm{f}}\,\bar{I}_1 + \left(1 - 3\kappa_{\mathrm{f}}\right)\mathbf{f}_0 \cdot \bar{\mathbf{C}}\mathbf{f}_0$$

For a list of material parameters, see Table A1.

## Appendix B: Kinematic growth

The model of kinematic growth according to Rodriguez et al. (1994) describes growth as alterations in the shape and size of an unloaded body through the inelastic growth deformation gradient $\mathbf{F}_{\mathrm{g}}$.

These stress-free changes, arising from mass addition or loss, induce an intermediate, incompatible configuration. Restoring geometric compatibility is then achieved through the elastic deformation gradient $\mathbf{F}_{\mathrm{e}}$, which organises the volumetric elements into an unloaded body, consequently generating residual stresses. The overall deformation gradient is then expressed as:

$$\mathbf{F} = \mathbf{F}_{\mathrm{e}}\mathbf{F}_{\mathrm{g}}. \tag{9}$$

This approach requires two distinct constitutive relationships: a strain–energy function $\Psi(\mathbf{F}_{\mathrm{e}}) = \Psi(\mathbf{C}_{\mathrm{e}})$ (see eqn (8)) to model the hyperelastic material behaviour and an evolution equation for the growth tensor.

To model transverse fibre growth due to chronic cardiomyocyte thickening, a growth multiplier, $\vartheta^\perp = \sqrt{\det\mathbf{F}_{\mathrm{g}}}$, is introduced to characterise a parallel deposition of sarcomeres at the molecular level. The formulation proposed by Genet et al. (2016) constructs the growth tensor as a rank-one update of the unity tensor, scaled by the growth multiplier, within the plane perpendicular to the fibre direction $\mathbf{f}_0$. This results in:

$$\mathbf{F}_{\mathrm{g}} = \vartheta^\perp\,I + \left(1 - \vartheta^\perp\right)\mathbf{f}_0 \otimes \mathbf{f}_0$$

Using eqn (9) an explicit expression for the elastic deformation tensor can be derived:

$F_{\mathrm{e}} = F + \frac{1-\vartheta^\parallel}{\vartheta^\parallel}f \otimes f_0.$

To simulate longitudinal fibre growth attributed to chronic cardiomyocyte lengthening, a scalar-valued growth multiplier, $\vartheta^\parallel = \det\mathbf{F}_{\mathrm{g}}$, is introduced to represent the sequential deposition of sarcomeres at the molecular level. This results in the formulation of the growth tensor for longitudinal fibre growth, achieved through a rank-one update of the unity tensor along the fibre direction $\mathbf{f}_0$:

$$\mathbf{F}_{\mathrm{g}} = \mathbf{I} + \left(\vartheta^\parallel - 1\right)\mathbf{f}_0 \otimes \mathbf{f}_0.$$

Again, by using eqn (9), an explicit expression for the elastic deformation gradient can be derived:

$$\mathbf{F}_{\mathrm{e}} = \mathbf{F} + \frac{1 - \vartheta^\parallel}{\vartheta^\parallel}\mathbf{f} \otimes \mathbf{f}_0.$$

As in (Genet et al., 2016), we assumed stretch-driven growth kinetics for both transverse and longitudinal growth and defined the growth multiplier as:

$$\dot{\vartheta}^\perp = \frac{1}{\tau_{\mathrm{type}}^\perp}\,\langle\lambda - \lambda_{\mathrm{crit}}\rangle, \quad \dot{\vartheta}^\parallel = \frac{1}{\tau^\parallel}\,\langle\lambda - \lambda_{\mathrm{crit}}\rangle$$

where $\tau^\parallel$ and $\tau_{\mathrm{type}}^\perp$, type $\in$ {sym, asym}, are scaling parameters in time (see Table A1). $\langle\bullet\rangle$ denote Macaulay brackets to ensure growth activation only when the current fibre stretch $\lambda = (\mathbf{f}_0 \cdot F^\top \cdot F \cdot \mathbf{f}_0)^{1/2}$ exceeds the physiological limit $\lambda_{crit}$.

Maximal growth is constrained by a sigmoid function, which serves as a mathematical model to represent the growth dynamics of the system. The sigmoid function is characterised by an S-shaped curve that captures the initial rapid growth phase, followed by a deceleration as the growth approaches a maximum limit. This

**Table A1. Input parameters for the 3D PDE model**

| Parameter | Value | Unit | Description |
|---|---|---|---|
| **Passive tissue – ventricles** | | | |
| $\kappa$ | 650 | kPa | Bulk modulus |
| $a$ (LV) | 2.72 | kPa | Stress-like parameter for isotropic contribution |
| $a$ (RV) | 10.88 | kPa | Stress-like parameter for isotropic contribution |
| $b$ | 21.75 | [-] | Dimensionless parameter for isotropic contribution |
| $a_f$ (LV) | 0.49 | kPa | Stress-like parameter for anisotropic contribution |
| $a_f$ (RV) | 1.96 | kPa | Stress-like parameter for anisotropic contribution |
| $b_f$ | 90.1 | — | Dimensionless parameter for anisotropic contribution |
| $\kappa_f$ | 0.08 | — | Dispersion parameter |
| **Passive tissue – atria** | | | |
| $\kappa$ | 650 | kPa | Bulk modulus |
| $a$ | 11.68 | kPa | Stress-like parameter for isotropic contribution |
| $b$ | 5.6 | — | Dimensionless parameter for isotropic contribution |
| $a_f$ | 23.68 | kPa | Stress-like parameter for anisotropic contribution |
| $b_f$ | 17.97 | — | Dimensionless parameter for anisotropic contribution |
| $\kappa_f$ | 0.17 | — | Dispersion parameter |
| **Kinematic growth** | | | |
| $p$inflate | 5.0 | mmHg | LV pressure used for passive inflation experiments |
| $p$overload | 10.0 | mmHg | LV pressure used for kinematic growth experiments |
| $\tau^{\parallel}$ | 0.4 | — | Eccentric growth scaling parameter |
| $\tau^{\perp}_{sym}$ | 0.4 | — | Symmetric concentric growth scaling parameter |
| $\tau^{\perp}_{asym}$ | 0.3 | — | Asymmetric concentric growth scaling parameter |
| $t$increase | 0.1 | — | Time interval where pressure increases from 0 to $p_{overload}$ |
| $t$growth | 0.6 | — | Time interval where pressure is constant at $p_{overload}$ |
| $t$decrease | 0.1 | — | Time interval where pressure decreases from $p_{overload}$ to 0 |
| **Boundary conditions** | | | |
| $k^{\parallel}_{epi}$ | 0.005 | kPa/µm | Epicardial spring stiffness for eccentric growth |
| $k^{\perp}_{epi,sym}$ | 0.01 | kPa/µm | Epicardial spring stiffness for symmetric concentric growth |
| $k^{\perp}_{epi,asym}$ | 0.3 | kPa/µm | Epicardial spring stiffness for asymmetric concentric growth |
| $k$vessel | 0.05 | kPa/µm | Spring stiffness applied to vessel rings |
| $k$endo | 0.005 | kPa/µm | Spring stiffness applied to the endocardium of the atria and the right ventricle |
| **Electrophysiology** | | | |
| $(v_l, v_t, v_n)$ | (0.6, 0.4, 0.2) | m/s | Conduction velocities in the ventricles |
| $(v_f, v_s, v_n)$ | (1.2, 0.8, 0.4) | m/s | Fast conducting layer velocities |
| $(\sigma il, \sigma it, \sigma in)$ | (0.34, 0.6, 0.6) | S/m | Intracellular conductivities |
| $(\sigma el, \sigma et, \sigma en)$ | (0.12, 0.8, 0.8) | S/m | Extracellular conductivities |
| $(\sigma bp, \sigma torso)$ | (0.7, 0.22) | S/m | Conductivities in blood pools and the rest of the torso |

behaviour is typical in biological systems, where growth is constrained by various biological and physiological factors. Mathematically, the sigmoid function is expressed as:

$$G\left(t\right) = \frac{2}{1 + \exp\left(\text{vol}_{elem} - \text{vol}_{elem,mean}\right)}$$

where $\text{vol}_{elem}$ is the current volume of the element and $\text{vol}_{elem,mean}$ is the mean volume over all elements in the mesh.

## Appendix C: Universal ventricular coordinates

For the basic understanding of the UVC approach, the reader should refer to the existing literature (Bayer et al., 2018).

In our algorithm, two key metrics were defined to parametrise anatomical locations based on the principles of the UVC method. These metrics were determined using two axes: the longitudinal axis of the ventricle and the radial axis around the longitudinal axis.

Unlike the original implementation of UVC generation, which defined the long axis of the ventricles as a straight

line extending from the apex to the base, a more adaptable approach was employed. The long axis in our method was determined by the centreline of the ventricle, which could vary based on the specific geometry of the heart. This flexibility was crucial, especially in cases of highly progressed concentric hypertrophy where the centreline of the ventricle might exhibit a curve-like shape rather than a straight line. By utilising the centreline as the long axis, our algorithm accommodated diverse ventricular geometries, ensuring accurate parametrisation and mapping of anatomical locations across different cardiac models.

The sub-endocardial layer was parametrised by assigning positional values to each point on the surface (see Fig. A1). These parameters were determined by measuring the position of each point along the longitudinal and circumferential axes. The longitudinal axis extended from the centre of the ventricle at the basal level, where the longitudinal coordinate ($L$) was 0, to the apex, where $L$ was 1. This axis allowed for the description of points along the long axis. The longitudinal position of each point on the sub-endocardial layer relative to the centreline was determined using this axis. The circumferential axis characterised the points around the longitudinal axis by assigning the polar coordinate $\theta$. $\theta$ was the angle measured in radians, beginning from 0 at the junction of a line connecting the centres of the LV and RV at the basal level and the LV lateral wall. It increased in a counter-clockwise direction around the circle encompassing the endocardial layer, reaching $2\pi$ after completing a full revolution.

## Appendix D: LVH and EP remodelling

Beyond structural adaptation, LVH significantly alters cardiac EP by prolonging action potential duration, particularly during repolarisation, and decreasing conduction velocity (CV) (Jin et al., 2010; Sung et al., 2003; Tomaselli & Marbán, 1999). Additionally, LVH reduces conduction anisotropy, by increasing side-to-side connections between ventricular myocytes (Carey et al., 2001), a common side-effect of fibrosis.

Studies indicate that LVH is associated with a slowing of CV with the slowing being more pronounced in the fibre direction where conduction is naturally faster (Akar et al., 2004; Bacharova et al., 2010; Wiegerinck et al., 2006). Some research suggests that this decrease in CV may be linked to an increase in intracellular resistance ($R_i$) (Botchway, 2003; Wiegerinck et al., 2006). Generally, one might expect that an increase in cell size would accelerate CV, but evidence shows that despite hypertrophy-induced cell enlargement, longitudinal CV actually decreases. This discrepancy can be explained by the fact that CV is influenced by both conduction speed within cells and between cells (Dhein et al., 2014). Beyond $R_i$, gap junction (GJ) resistance plays a significant role in CV. Given that GJ resistance is higher than cytoplasmic resistance, the regulation of GJ conductance plays a crucial role in determining CV (Bukauskas & Verselis, 2004). For instance, increasing the number of GJs can alter CV. As hypertrophic myocytes replicate and the number of GJs increases, LVH becomes a cardiac event that notably alters CV.

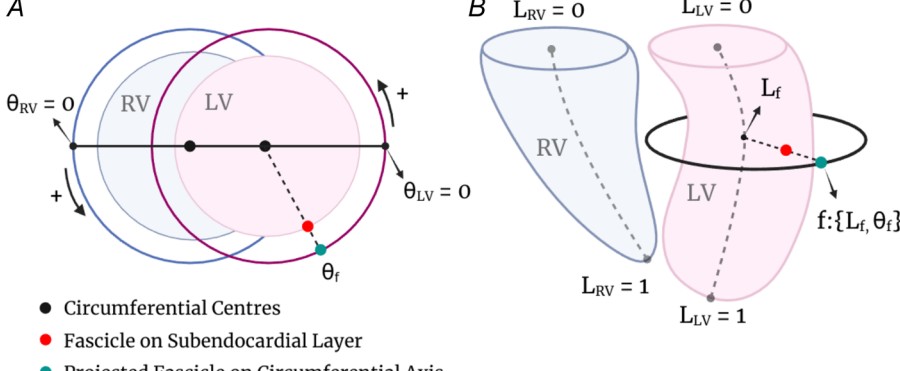

**Figure A1. Parametrisation of points on the sub-endocardial layer of the ventricles**
*A*, parametrisation along the circumferential axis, with polar coordinates $\theta$ assigned to each point. The angle $\theta$, measured in radians, starts from 0 at the junction of the line connecting the centres of the left and right ventricles at the basal level and the left ventricular lateral wall and increases counter-clockwise. *B*, parametrisation along the long axis of the ventricles. The longitudinal axis extends from the centre of the ventricle at the basal level (where $L$ = 0) to the apex (where $L$ = 1). Each point on the sub-endocardial layer is defined by its position along both the circumferential and longitudinal axes.

The decrease in CV, along with changes in conduction anisotropy, can lead to various abnormalities, including arrhythmias (Cooklin et al., 1997, 1998; Peters, 1996). In clinical practice, ventricular arrhythmias are directly linked to LVH (Wolk, 2000), with a particularly pronounced connection observed in patients with hypertension (Cooper & Liao, 1991). The slowed conduction within hypertrophied myocardium is a critical factor in the heightened incidence of arrhythmias. This slowdown extends conduction time along existing re-entrant pathways, thereby increasing the likelihood of arrhythmic events (Wolk, 2000).

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

## Additional information

### Data availability statement

Meshes of four-chamber models of left ventricular hypertrophy will be made available on Zenodo upon acceptance of this article. Due to the extensive storage requirements (over 1TB per case), simulation data cannot be hosted on a public repository but will be provided upon reasonable request for non-commercial use. For further inquiries, please contact the corresponding author.

### Competing interests

The authors declare that they have no competing interests.

### Author contributions

M.K.: Conceptualization, Data curation, Formal analysis, Investigation, Methodology, Software, Validation, Visualization, Writing – original draft, Writing – review & editing. K.G.: Formal analysis, Investigation, Software, Writing – review & editing. M.A.F.G.: Software, Writing – review & editing. A.J.P.: Software, Writing – review & editing. G.P.: Software, Funding Acquisition, Writing – review & editing. C.M.A.: Conceptualization, Data curation, Formal analysis, Funding Acquisition, Investigation, Methodology, Project Administration, Resources, Software, Supervision, Validation, Writing – original draft, Writing – review & editing. All authors approved the final version of the manuscript submitted for publication and agree to be accountable for all aspects of the work. All persons designated as authors qualify for authorship, and all those who qualify for authorship are listed.

## Funding

This research was funded in whole or in part by the Austrian Science Fund (FWF) grant DOIs 10.55776/P37063 and 10.55776/I4652 to C.M.A, and grant 10.55776/I6540 to G.P. For open access purposes, the author has applied a CC BY public copyright license to any author-accepted manuscript version arising from this submission.

## Keywords

biomechanics, cardiac electrophysiology, cardiac function, computer modelling, hypertrophy

## Supporting information

Additional supporting information can be found online in the Supporting Information section at the end of the HTML view of the article. Supporting information files available:

**Peer Review History**

