## [Peer Review History · The Journal of Physiology]

Computational Modelling of the Impact of Anatomical Changes on ECGs in Left Ventricular Hypertrophy

Christoph M Augustin, Mohammadreza Kariman, Matthias A.F. Gsell, Anton J Prassl, Karli Gillette, and Gernot Plank
DOI: 10.1113/JP287954

Corresponding author(s): Christoph Augustin (christoph.augustin@medunigraz.at)

The following individual(s) involved in review of this submission have agreed to reveal their identity: Jonathan F Wenk (Referee #1); Vicky Wang (Referee #2)

Review Timeline:

Submission Date:	31-Oct-2024
Editorial Decision:	02-Jan-2025
Revision Received:	26-Jun-2025
Accepted:	17-Jul-2025

Senior Editor: Eleonora Grandi

Reviewing Editor: Brian Delisle

Transaction Report:

Dear Dr Augustin,

Re: JP-RP-2024-287954 "Computational Modelling of the Impact of Anatomical Changes on ECGs in Left Ventricular Hypertrophy" by Christoph M Augustin, Mohammadreza Kariman, Matthias A.F. Gsell, Anton J Prassl, Karli Gillette, and Gernot Plank

Thank you for submitting your manuscript to The Journal of Physiology. It has been assessed by a Reviewing Editor and by 2 expert referees and we are pleased to tell you that it is potentially acceptable for publication following satisfactory major revision.

REVISION CHECKLIST:

We look forward to receiving your revised submission.

Yours sincerely,

Natalia Trayanova
Senior Editor
The Journal of Physiology

REQUIRED ITEMS

- Author photo and profile. First or joint first authors are asked to provide a short biography (no more than 100 words for one author or 150 words in total for joint first authors) and a portrait photograph. These should be uploaded and clearly labelled together in a Word document with the revised version of the manuscript. See Information for Authors for further details.
- You must start the Methods section with a paragraph headed Ethical Approval. If experiments were conducted on humans, confirmation that informed consent was obtained, preferably in writing, that the studies conformed to the standards set by the latest revision of the Declaration of Helsinki and that the procedures were approved by a properly constituted ethics committee, which should be named, must be included in the article file. If the research study was registered (clause 35 of the Declaration of Helsinki), the registration database should be indicated, otherwise the lack of registration should be noted as an exception (e.g. The study conformed to the standards set by the Declaration of Helsinki, except for registration in a database). For further information see: <https://physoc.onlinelibrary.wiley.com/hub/human-experiments>.
- The reference list must be in alphabetical order, rather than numbered, to comply with our Journal format.
- Your manuscript must include a complete Additional Information section, including competing interests; funding; author contributions and acknowledgements.
- Please upload separate high-quality figure files via the submission form.
- Please ensure that any tables are editable and in Word format, and wherever possible, embedded in the article file itself.
- Please ensure that the Article File you upload is a Word file.
- Please include an Abstract Figure file, as well as the Figure Legend text within the main article file. The Abstract Figure is a piece of artwork designed to give readers an immediate understanding of the research and should summarise the main conclusions. If possible, the image should be easily 'readable' from left to right or top to bottom. It should show the physiological relevance of the manuscript so readers can assess the importance and content of its findings. Abstract Figures should not merely recapitulate other figures in the manuscript. Please try to keep the diagram as simple as possible and without superfluous information that may distract from the main conclusion(s). Abstract Figures must be provided by authors

no later than the revised manuscript stage and should be uploaded as a separate file during online submission labelled as File Type 'Abstract Figure'. Please also ensure that you include the figure legend in the main article file. All Abstract Figures should be created using BioRender. Authors should use The Journal's premium BioRender account to export high-resolution images. Details on how to use and access the premium account are included as part of this email.

EDITOR COMMENTS

Reviewing Editor:

Two experts in the field reviewed the manuscript. Both referees found that the article was interesting and had the potential to be impactful. Each referee identified several issues that limit the impact of the study in its current form.

Some of the data is derived from human measurements from a previous publication (reference 39). This might require more detail in the manuscript. See P4 Methods section "A detailed anatomical model of a healthy human heart, along with a reconstruction of the subject's torso, was generated using magnetic resonance (MR) images from a 45-year-old male subject. This MRI study received approval from the Ethical Review Board of the Medical University of Graz (EKNr: 24-126 ex 11/12), and participants provided written informed consent. For further details on image acquisition, we refer to Gillette et al. [39]."

Please also see 'Required Items' above.

REFEREE COMMENTS

Referee #1:

The authors have constructed an anatomically accurate FE model of a healthy human heart and then used a growth algorithm to induce various levels of eccentric and concentric hypertrophy. These models were then simulated to assess alterations in ECG signals, based solely on anatomical changes. The work is interesting, and the following comments are submitted for the author's consideration.

1. In the methods section, the helical fiber angles are assigned to be -60 to 60 degrees, with no mention of the transverse angles. This reviewer assumes they are 0 degrees, but wonder if those would have an effect on the signal, as they would affect the direction of conduction.
2. The most significant comment is related to the description of the kinematic growth procedure. Please provide more specific details about this process. Was the growth tensor assigned at each integration point a priori, or was a load applied to the model and the growth tensor was solved for as part of a boundary value problem. It is unclear how the final 20 states were generated. This is particularly important for the concentric cases, which seem to show a much larger septum (Figure 2). In the case of hypertension or aortic stenosis (typical pressure overload cases) the increase in wall thickness is more uniform circumferentially about the LV. A thick septum is more consistent with HCM (which is related to sarcomeric mutations). Considering that the wall thickness in the septum and free wall is fairly similar in the baseline model, it is important to describe how the final growth states were generated.
3. Regarding the positioning of the heart in the abdominal mesh, the eccentric case with 100% growth (and possibly for other cases, which are not shown) appears to get very close to the surface of the chest. Presumably, there would be ribs in that location and the heart would not be able to extend that close. The authors describe their approach for positioning, but could this case be errant for the larger growth cases. What does this look like in a real patient with dilation?
4. Was the scaling factor in section 2.7 based on this specific anatomy? Different values were used in the referenced prior works.

5. In section 2.9, the authors may want to define i as the lead number, j as the patient number, and the D 's.
6. Related to comment 2, it might be worth looking at a set of simulations where the concentric hypertrophy is more "concentric". Could the geometry used in this study (with the thicker septum) affect the results presented? Could that be a part of the inconsistencies that the authors report with those results? Also, could this related to the ECG criteria presented in section 3.5, where none of them predicted LVH for the concentric cases.
7. Since Figures 4 and 5 focus on the 50% and 100% hypertrophy cases, it might be worth altering Figure 2 to show those geometries compared to the baseline model. This reviewer would certainly be interested to see the geometry of the 100% case in particular, since the septum is so large even for just the 40% case.
8. For Figure 7, would it be possible to separate the results into the eccentric and concentric cases? This would show the distinction between the lead sensitivities for these two cases. Right now, they are blended together.
9. In section 4.2, there is a typo when describing how hypertrophy develops. It is the parallel and serial deposition of sarcomeres within the myocytes. Not the replication of myocytes.
10. The reviewer appreciates the mention in the limitations regarding alterations in fiber orientation and the development of fibrosis. These would be significant factors in a chronic setting.

Referee #2:

This manuscript reported a systematic computational modelling study which investigated the effect of wall thickness changes on the behaviour of ECGs. This study was largely motivated by the need to better understand the alteration of ECGs for subjects with left ventricular hypertrophy. The design of this *in silico* study is really structured and comprehensive which is one of the major strengths of this work. The manuscript was also professionally prepared with sufficient attention given to each section. The detailed analyses of the results of each type of alteration provide some levels of insight, and its scope fits with that of the journal. Therefore, the reviewer thinks the standard held by this manuscript does meet the high-quality standard held by the journal.

Strengths:

1. The writing style of this manuscript is commendable, and the level of detail given by the authors is also impressive.
2. The design of the simulation scenarios is also quite comprehensive which makes this study a very relevant mechanistic study albeit it was an *in silico* analysis.

Please see below some specific comments for the authors to consider during the revisions.

1. The simulation of the concentric growth appeared to have resulted in significant septal wall thickening as Figure 2 indicates. Did the author anticipate such an effect?
2. The study adopted the rule-based fiber field and it is known that both growth and EP modelling are dependent on the assumption of the microstructural orientation. Can the authors discuss this aspect?

3. Figure 3 demonstrates the changes in the isochronal maps of the ventricular depolarisation under different growth settings. Upon closer inspection, it seemed that the geometry over which the map was overlaid corresponded to the same geometry. Therefore, the reviewer recommends the authors to show changes in the isochronal map to better highlight the effects.

END OF COMMENTS

Response to Referees

We, the authors, would like to thank the referees for their time invested in reading our manuscript and for providing helpful reviews and suggestions for improvement. We have included detailed responses to each comment below and have made requested changes to the manuscript. To assist with the review, we have included a version of the manuscript that highlights changes corresponding to input of both reviewers in green, of Reviewer # 1 in blue, and of Reviewer # 2 in red.

RESPONSE

Reviewer # 1: The authors have constructed an anatomically accurate FE model of a healthy human heart and then used a growth algorithm to induce various levels of eccentric and concentric hypertrophy. These models were then simulated to assess alterations in ECG signals, based solely on anatomical changes. The work is interesting, and the following comments are submitted for the author's consideration.

REFEREE

We thank the reviewer for the overall positive view on our study. We took all remarks into account and also revised manuscript, see below.

RESPONSE

Major remarks:

REFEREE

1. In the methods section, the helical fiber angles are assigned to be -60 to 60 degrees, with no mention of the transverse angles. This reviewer assumes they are 0 degrees, but wonder if those would have an effect on the signal, as they would affect the direction of conduction.

REFEREE

We thank the reviewer for this comment, as we had not previously provided transverse angles. We have now clarified the choice of longitudinal and transverse fibre directions in the document:

RESPONSE

"Fibre architecture within the ventricles was generated using a rule-based method as described by Bayer et al. 2012, with longitudinal fiber angles rotating from 60° to -60° and transverse fibre angles rotating from -65° to 25° , from the endocardial to the epicardial surface."

This fiber angle configuration follows the approach of Gillette et al. 2021, which serves as the baseline case for our investigation. Of course, variations in longitudinal and transverse fibre orientations would have an effect on the signal, but they would also influence the fitting procedure to the clinically measured ECG signal, which is not the main focus of this paper. We have included a sentence in the limitations sections to address the choice of fibre orientation. See also our response to Remark 2 of Reviewer #2.

2. The most significant comment is related to the description of the kinematic growth procedure. Please provide more specific details about this process. Was the growth tensor assigned a priori, or was a load applied to the model and the growth tensor was solved for as part of a boundary value problem. It is unclear how the final 20 states were generated. This is particularly important for the concentric cases, which seem to show a much larger septum (Figure 2). In the case of hypertension or aortic stenosis (typical pressure overload cases) the increase in wall thickness is more uniform circumferentially about the LV. A thick septum is more consistent with HCM (which is related to sarcomeric mutations). Considering that the wall thickness in the septum and free wall is fairly similar in the baseline model, it is important to describe how the final growth states were generated.

REFEREE

The asymmetry observed in the concentric growth case, also noted by Reviewer #2 (see our response to Remark 1), prompted a thorough re-evaluation. In the original setup, relatively stiff epicardial boundary conditions caused elevated septal strain due to less constraint on the septum deforming into the right ventricle; ultimately leading to pronounced thickening of the septum. We repeated the

RESPONSE

simulations with reduced epicardial boundary stiffness, allowing more deformation of the LV free wall. This yielded a more symmetric growth pattern. We now include both symmetric and asymmetric cases in the manuscript, as both are clinically relevant. The manuscript, tables, and figures were updated accordingly. The overall conclusions remain unchanged, as both scenarios support our main findings, see also our response to Remark 6.

Additionally, we tried to clarify the description of the kinematic growth procedure and shifted parts of the "Appendix B", where the model was described in more detail, to the main manuscript. We chose not to move the equations from the appendix into the main manuscript, as the Journal of Physiology is not heavily focused on mathematical formulations. Instead, we maintained a concise presentation in the main text while providing all relevant equations and technical details in the appendix for interested readers.

3. Regarding the positioning of the heart in the abdominal mesh, the eccentric case with 100% growth (and possibly for other cases, which are not shown) appears to get very close to the surface of the chest. Presumably, there would be ribs in that location and the heart would not be able to extend that close. The authors describe their approach for positioning, but could this case be errant for the larger growth cases. What does this look like in a real patient with dilation? REFEREE

As described in Section 2.7, the lungs and ribs were omitted from the computational torso model to accommodate extreme hypertrophy cases (e.g., 80-100% growth), which represent rare pathological scenarios. In the real scenario, the rib cage would indeed impose anatomical constraints on the heart's position and growth. However, the current model was designed to explore theoretical limits and understand the physiological implications under extreme conditions. Incorporating rib structures into future models could provide additional insights, but for this study, our simplified approach allows a focus on the cardiac growth dynamics without external constraints. RESPONSE

4. Was the scaling factor in section 2.7 based on this specific anatomy? Different values were used in the referenced prior works. REFEREE

The removal of the rib cage and lungs necessitated the use of a different scaling factor to align the simulated ECG with measured ECG data. This adjustment accounts for the anatomical simplifications in our model and ensures the validity of the simulated signals. While prior works used different values, the chosen scaling factor in this study was tailored specifically to compensate for the anatomical modifications in our model. We added a sentence in the manuscript to clarify this issue. RESPONSE

5. In section 2.9, the authors may want to define i as the lead number, j as the patient number, and the D 's. REFEREE

We thank the reviewer for identifying the missing definitions. These have now been added to the manuscript. RESPONSE

6. Related to comment 2, it might be worth looking at a set of simulations where the concentric hypertrophy is more "concentric". Could the geometry used in this study (with the thicker septum) affect the results presented? Could that be a part of the inconsistencies that the authors report with those results? Also, could this related to the ECG criteria presented in section 3.5, where none of them predicted LVH for the concentric cases. REFEREE

See also our response to Remark 2. The simulations for both eccentric and concentric hypertrophy cases were carefully analysed, focusing on the spatiotemporal distribution of transmembrane potential RESPONSE

in the ventricles. As discussed in Section 4.2, the increased septal wall thickness primarily delays RV activation due to its influence on septal conduction (as shown in Figure 4). However, LV activation is mainly driven by the anterior, septal, and posterior fascicles of the LBB. Importantly, the septal fascicle, being located in the sub-endocardial layer, activates the LV endocardium independently of septal thickness. Thus, the degree of septal thickening does not directly affect LV activation timing.

The ECG leads most influenced in concentric hypertrophy (aVL, II, and V4) primarily reflect LV activity, indicating that the conclusions drawn from this study were not significantly affected by the increased septal thickness.

It is also worth noting that established ECG criteria for LVH detection are not classified by hypertrophy mode, which could contribute to the failure to detect LVH in our extreme concentric hypertrophy cases. These extreme scenarios were intentionally included to investigate correlations between anatomical changes and ECG morphology across a wide spectrum of growth.

Finally, this study intentionally excluded EP remodelling to isolate the effects of anatomical changes on ECG morphology. However, it is acknowledged that in realistic scenarios, such as complex HCM cases, EP remodelling may play a critical role and should be considered in future investigations.

7. Since Figures 4 and 5 focus on the 50% and 100% hypertrophy cases, it might be work altering Figure 2 to show those geometries compared to the baseline model. This reviewer would certainly be interested to see the geometry of the 100% case in particular, since the septum is so large even for just the 40% case. REFEEER

We changed Figure 2 to include the 50% and 100% cases for eccentric as well as symmetric and asymmetric concentric cases. The septum for the asymmetric concentric case is indeed large, however, extreme growth patterns like that are still observed, see, e.g., Basso et al (2021) doi: 10.1007/s00428-021-03038-0. RESPONSE

8. For Figure 7, would it be possible to separate the results into the eccentric and concentric cases? This would show the distinction between the lead sensitivities for these two cases. Right now, they are blended together. REFEEER

Unfortunately, the Physionet dataset does not provide a classification of hypertrophy types into eccentric and concentric categories, but instead only includes data labeled as LVH. As a result, it was not possible to separate the results into eccentric and concentric hypertrophy cases. Future studies could explore datasets with more detailed hypertrophy classifications to enable such a distinction. RESPONSE

9. In section 4.2, there is a typo when describing how hypertrophy develops. It is the parallel and serial deposition of sarcomeres within the myocytes. Not the replication of myocytes. REFEEER

We changed that according to the reviewers suggestion. RESPONSE

10. The reviewer appreciates the mention in the limitations regarding alterations in fiber orientation and the development of fibrosis. These would be significant factors in a chronic setting. REFEEER

Thank you! We added a further sentence in the limitations sections regarding fiber orientations; see also Remark 1. RESPONSE

Reviewer # 2: This manuscript reported a systematic computational modelling study which investigated the effect of wall thickness changes on the behaviour of ECGs. This study was largely motivated by the need to better understand the alteration of ECGs for subjects with left ventricular hypertrophy. The design of this in silico study is really structured and comprehensive which is one of the major strengths of this work. The manuscript was also professionally prepared with sufficient attention given to each section. The detailed analyses of the results of each type of alteration provide some levels of insight, and its scope fits with that of the journal. Therefore, the reviewer thinks the standard held by this manuscript does meet the high-quality standard held by the journal.

REFeree

Strengths:

1. The writing style of this manuscript is commendable, and the level of detail given by the authors is also impressive.
2. The design of the simulation scenarios is also quite comprehensive which makes this study a very relevant mechanistic study albeit it was an in-silico analysis.

We thank the reviewer for this positive evaluation of our study and we are grateful for the recognition that our work meets the high-quality standard of the Journal of Physiology. All remarks raised have been carefully taken into account, see below.

RESPONSE

Specific comments:

REFeree

1. The simulation of the concentric growth appeared to have resulted in significant septal wall thickening as Figure 2 indicates. Did the author anticipate such an effect?

REFeree

The asymmetry observed in the concentric growth case was also noted by Reviewer #1 (see our response to Remark 2). We thoroughly investigated the origin of the pronounced septal wall thickening and repeated all concentric growth simulations with slightly adjusted model parameters: in the initial setup, we applied relatively stiff boundary conditions on the epicardial surface of the heart, including the left ventricular free wall. This resulted in increased strain in the septum, which was less constrained by deforming into the right ventricle. The elevated strain levels in the septum led to more growth compared to the LV free wall. In the revised setup, we reduced the stiffness of the epicardial boundary conditions, allowing for greater deformation of the LV free wall. This resulted in increased growth in the free wall and, consequently, a more symmetric concentric growth pattern. We have included this symmetric case in our analysis, while retaining the original asymmetric case in the manuscript, as asymmetric concentric growth patterns are also commonly observed in patients. As a result of this change, we updated most of the figures and revised substantial parts of the manuscript. Importantly, the overall conclusions of the paper remain unchanged, as the findings from both the symmetric and asymmetric concentric growth cases were largely consistent.

RESPONSE

2. The study adopted the rule-based fiber field and it is known that both growth and EP modelling are dependent on the assumption of the microstructural orientation. Can the authors discuss this aspect?

REFeree

We thank the reviewer for this comment and we fully agree that growth and EP modelling are highly dependent on fibre fields. We adopted the fibre angle configuration from Gillette et al. (2021), where the fitted EP model serves as the baseline case for our investigation. Since myocardial fibre orientations cannot be measured in-vivo with sufficient resolution, rule-based methods currently represent the state-of-the-art for assigning fibre architecture in computational cardiac models. While variations in longitudinal and transverse fibre orientations would indeed affect growth kinematics and the resulting electrical signal, they would also influence the fitting procedure to the clinically recorded ECG. This

RESPONSE

fitting was done in Gillette et al. (2021) but is beyond the scope of our manuscript. We have included a note in the limitations section to discuss this shortcoming. See also Remark 1 and 10 of Reviewer #1.

3. Figure 3 demonstrates the changes in the isochronal maps of the ventricular depolarisation under different growth settings. Upon closer inspection, it seemed that the geometry over which the map was overlaid corresponded to the same geometry. Therefore, the reviewer recommends the authors to show changes in the isochronal map to better highlight the effects. REFeree

We thank the reviewer for the careful reading of our manuscript. We would like to clarify that the geometries shown in Figure 3 are indeed different and reflect the respective growth settings applied in each case. In the concentric hypertrophy scenarios, the outer shape of the ventricles did not change markedly; however, there is a clear increase in overall size when comparing the control case to, for example, the 100% eccentric growth case. RESPONSE

While the anatomical appearance may seem similar for many of the cases, the underlying geometrical differences introduced by the growth parameters are present and do influence the resulting isochronal maps. We have aimed to highlight these effects using white arrows in Figure 3 in the 50% and 100% growth cases and through the corresponding discussion in the manuscript.

Dear Professor Augustin,

Re: JP-RP-2025-287954R1 "Computational Modelling of the Impact of Anatomical Changes on ECGs in Left Ventricular Hypertrophy" by Christoph M Augustin, Mohammadreza Kariman, Matthias A.F. Gsell, Anton J Prassl, Karli Gillette, and Gernot Plank

We are pleased to tell you that your paper has been accepted for publication in The Journal of Physiology.

Yours sincerely,

Eleonora Grandi
Senior Editor
The Journal of Physiology

If you would like to receive our 'Research Roundup', a monthly newsletter highlighting the cutting-edge research published in The Physiological Society's family of journals (The Journal of Physiology, Experimental Physiology, Physiological Reports, The Journal of Nutritional Physiology and The Journal of Precision Medicine: Health and Disease), please click this link, fill in your name and email address and select 'Research Roundup':

<https://www.physoc.org/journals-and-media/membernews>

- You can help your research get the attention it deserves! Check out Wiley's free Promotion Guide for best-practice recommendations for promoting your work at: www.wileyauthors.com/eoo/guide. You can learn more about Wiley Editing Services which offers professional video, design, and writing services to create shareable video abstracts, infographics, conference posters, lay summaries, and research news stories for your research at: www.wileyauthors.com/eoo/promotion.

EDITOR COMMENTS

Reviewing Editor:

The revised manuscript has addressed the referee's previous concerns, and it is expected to have a significant impact on the field.

Senior Editor:

Thank you for a thoughtful revision. I concur with the reviewing editor's assessment.

REFEREE COMMENTS

Referee #1:

The authors have addressed all of my comments. Thank you.

Referee #2:

The authors have sufficiently addressed the comments and questions raised in the original review.